# ETTA: Elucidating the Design Space of Text-to-Audio Models

**Sang-gil Lee** [* 1] **Zhifeng Kong** [* 1] **Arushi Goel** [1] **Sungwon Kim** [1] **Rafael Valle** [1] **Bryan Catanzaro** [1]

## Abstract

Recent years have seen significant progress in Text-To-Audio (TTA) synthesis, enabling users to enrich their creative workflows with synthetic audio generated from natural language prompts. Despite this progress, the effects of data, model architecture, training objective functions, and sampling strategies on target benchmarks are not well understood. With the purpose of providing a holistic understanding of the design space of TTA models, we set up a large-scale empirical experiment focused on diffusion and flow matching models. Our contributions include: 1) AF-Synthetic, a large dataset of high quality synthetic captions obtained from an audio understanding model; 2) a systematic comparison of different architectural, training, and inference design choices for TTA models; 3) an analysis of sampling methods and their Pareto curves with respect to generation quality and inference speed. We leverage the knowledge obtained from this extensive analysis to propose our best model dubbed Elucidated Text-To-Audio (ETTA). When evaluated on AudioCaps and MusicCaps, ETTA provides improvements over the baselines trained on publicly available data, while being competitive with models trained on proprietary data. Finally, we show ETTA's improved ability to generate creative audio following complex and imaginative captions – a task that is more challenging than current benchmarks[1].

## 1. Introduction

The design space of text-to-audio (TTA) models is complex, including a myriad of correlated factors. While our

research community has attempted to understand this design space and the contribution of each factor, drawing conclusions between experiments is challenging. Our goal in this work is not to explore novel model designs or methodologies. Instead, we aim to provide a holistic understanding of existing paradigms for building TTA models, to identify important aspects that allow for improving results, and to assess scalability with respect to data and model size.

In this paper, we aim to elucidate the design space of TTA model with respect to training data, model architecture and implementation, model capacity, training objective, and sampling methods during inference. In a controlled scenario and with a vast sweep over factors, we offer insights on the contribution of each factor. In addition to elucidating the design space of TTA models, our best configuration produces a model – namely Elucidated Text-to-Audio (ETTA) – that significantly improves over open-sourced baselines on both AudioCaps (Kim et al., 2019) and MusicCaps (Agostinelli et al., 2023) benchmarks with a single model.

Recent research has shown that scaling dataset size, combined with a careful data filtering strategy, can yield sizable improvements on benchmarks in other domains (Radford et al., 2019; Betker et al., 2023). Comparatively, the datasets used in TTA are generally much smaller, and their captions of varying quality, thus posing a challenge to scaling datasets (Liu et al., 2023b; Huang et al., 2023c). In order to circumvent these challenges, we introduce a large-scale and high-quality dataset of synthetic captions, and show that it is possible to leverage synthetic captions to obtain significant improvements.

While Transformers (Vaswani, 2017) have become the *de facto* architecture choice in many domains, sometimes their efficiency and stability, specially in larger models, are severely impaired by implementation details related to numerical precision and weight initialization. [2] We improve on several implementation details of the Diffusion Transformer (DiT) (Peebles & Xie, 2023) in the area of TTA generation, and provide insights on which details are important for improving benchmark scores.

In tandem, current trends have shown the benefits of scaling

---

[*]Equal contribution [1]NVIDIA. Correspondence to: Sang-gil Lee <sanggill@nvidia.com>, Zhifeng Kong <zkong@nvidia.com>, Rafael Valle <rafaelvalle@nvidia.com>.

*Proceedings of the $42^{nd}$ International Conference on Machine Learning*, Vancouver, Canada. PMLR 267, 2025. Copyright 2025 by the author(s).

[1]Demo: https://research.nvidia.com/labs/adlr/ETTA/, Code: https://github.com/NVIDIA/elucidated-text-to-audio

[2]E.g., see https://unsloth.ai/blog/gemma-bugs for the importance of implementation details.

model size (OpenAI, 2024; Chung et al., 2024; Radford et al., 2019), including better result on benchmarks and the appearance of emergent capabilities. While increasing capacity overall can yield improvements, it is important to strategically allocate capacity in a way that is Pareto optimal, maximizing scores and alleviating inference costs. In addition to increasing the decoder's capacity, the community has compared CLAP (Wu et al., 2023) and T5-based (Raffel et al., 2020; Chung et al., 2024) text encoders (Liu et al., 2023b; Ghosal et al., 2023; Liu et al., 2024), but the results seem mixed and strongly dependent on the data and decoder capacity at hand. We show in our experiments that, although improvements can be obtained by scaling model size, some strategies for increasing capacity yield better returns than others.

Finally, the diffusion model literature (Ho et al., 2020; Song et al., 2021) includes a wide range of training and sampling methods on the shelf (Kingma et al., 2021; Salimans & Ho, 2022; Lipman et al., 2022; Ho & Salimans, 2022; Karras et al., 2022; Tong et al., 2023; Karras et al., 2024). Through comprehensive experiments across various training objectives and sampling methods, we determine the most effective training method for our setting. In addition, we provide deeper insights into how to optimally select the sampling method for the best results by drawing Pareto curves across various evaluation metrics.

We summarize our contributions below:

- We introduce a large-scale and high-quality synthetic caption dataset called AF-Synthetic, and show that it can significantly improve text-to-audio generation quality on benchmarks.

- We ablate on major design choices in the text-to-audio space, and elucidate the importance of each component with respect to improving scores on benchmarks with an emphasis on data, architectural design, training objectives, and sampling methods.

- We introduce an improved implementation of diffusion transformer (DiT) for text-to-audio.

- We present ETTA, the *state-of-the-art* text-to-audio model trained on publicly available datasets. ETTA is also comparable with models trained on much larger proprietary data.

- We show ETTA's improved ability to generate creative audio following complex and imaginative captions.

## 2. Related Works

**Diffusion and Flow Matching Based Models**    Diffusion models (Ho et al., 2020; Song et al., 2021; Kong et al., 2021;

Kingma et al., 2021; Dhariwal & Nichol, 2021) are a type of deep generative models that learn the data distribution with optional conditions (e.g. text-to-X generation). They learn a reverse stochastic process that gradually transforms the Gaussian noise into clean data. The training objective of diffusion models is to predict the score function, i.e. the gradient of the log-likelihood with respect to data, via a neural network. Alternatively, some flow matching models predict the vector field related to the optimal transport between distributions (Lipman et al., 2022; Tong et al., 2023). These models can also be trained in the latent space (Rombach et al., 2022; Liu et al., 2023b) for better efficiency, scalability, and quality. Appendix B includes the mathematical details.

**Text-to-Audio Models**    There are two main streams of text-to-audio (TTA) models (including both audio and music generation) in the research community. One line of work uses diffusion and flow matching-based models. These works proposed numerous architectural and training designs for audio generation (Liu et al., 2023b; Ghosal et al., 2023; Huang et al., 2023c;a; Liu et al., 2024; Kong et al., 2024b; Xue et al., 2024; Haji-Ali et al., 2024; Hai et al., 2024; Vyas et al., 2023) and music generation (Melechovsky et al., 2023; Huang et al., 2023b; Evans et al., 2024a;b;c; Lam et al., 2024; Schneider et al., 2024; Lan et al., 2024; Li et al., 2024b;a; Fei et al., 2024). However, there is no systematic study on their design choices, and a main challenge is that the design space has too many variables to investigate. Our work falls in this category and aims at conducting the first systematic study on the design space of diffusion and flow matching based TTA models, and we choose to use the latest Stable Audio Open (Evans et al., 2024c) as our base model to investigate. Another line of research focuses on the language model approach and uses next token prediction to train a language model on discrete token representation of audio (Kreuk et al., 2022; Borsos et al., 2023; Agostinelli et al., 2023; Copet et al., 2024). These works are orthogonal to our study.

**Audio-Caption Datasets**    AudioSet (Gemmeke et al., 2017) pioneered large-scale audio-text dataset with labels for about 2M audio segments. AudioCaps (Kim et al., 2019) and MusicCaps (Agostinelli et al., 2023) are subsets of AudioSet with high-quality human-annotated captions. They are among the most common benchmarks for text-to-audio and text-to-music generation. With the rapid progress in large language models (LLMs) in recent years, LLM-enhanced audio-caption datasets such as WavCaps (Mei et al., 2024) and Laion-630K (Wu et al., 2023) were proposed, enabling large-scale audio-language models including TTA and other tasks. However, the captions can be noisy as the caption generation process does not depend on the audio signals. In the domain of TTA, recent works have used different collections of audio-caption pairs (mostly by

combining existing datasets) in order to train powerful TTA models. Examples include TangoPromptBank (Ghosal et al., 2023), AudioLDM (Liu et al., 2024), and Make-an-Audio (Huang et al., 2023c). However, these works mostly constitute combination and/or augmentation of existing data.

**Synthetic Data for Improved TTA**  Very recently, several concurrent works have studied using audio captioning models to generate synthetic captions of unlabeled audio. This leads to more accurate audio-caption pairs that could be used to train better TTA models. In detail, Sound-VECaps (Yuan et al., 2024) uses CogVLM (Wang et al., 2023) to generate visual descriptions and EnClap (Kim et al., 2024) to generate sound descriptions, and then use ChatGPT to condense into captions. This approach does not apply to audio data without video, and the captions may contain excessive visual information that does not exist in audio. Tango-AF is trained on AF-AudioSet (Kong et al., 2024b) generated with Audio Flamingo (Kong et al., 2024a). It has very high quality, but is very small in scale. GenAU (Haji-Ali et al., 2024) is a concurrent study to ours, trained on captions generated with AutoCap (Haji-Ali et al., 2024). All these studies demonstrate synthetic captions could lead to significant improvement of TTA generation quality. Inspired by these pioneering studies, we propose a large-scale synthetic dataset of captions leveraging an audio language model followed by filtering that ensures high quality captions.

## 3. Methodology

In Section 3.1, we introduce our method for building a large-scale, high-quality synthetic dataset used to train our TTA models. In Section 3.2, we describe our ETTA model, including architectural design, training objectives, and training methods of the variational autoencoder (VAE) and latent diffusion model (LDM). In Section 3.3, we describe the sampling algorithms that we will study in our experiments.

### 3.1. AF-Synthetic

Inspired by the recent success of synthetic captions in the text-to-image domain (Betker et al., 2023; Nguyen et al., 2024), we aim to build a large-scale and high-quality synthetic captions dataset for better text-to-audio models. While there are several in-the-wild datasets with paired text and audio data, they have certain limitations that we aim to overcome. Captions in WavCaps (Mei et al., 2024) and Laion-630K (Wu et al., 2023) are noisy because they are produced from text metadata only, not considering the actual audio. Sound-VECaps does not apply to audio data without video, and the captions may contain excessive visual information that does not exist in audio. AutoCap (Haji-Ali et al., 2024) and AF-AudioSet (Kong et al., 2024b) are closest to ours; AF-Synthetic constitutes scaling this approach.

We follow and improve the caption synthesis pipeline from AF-AudioSet. We use Audio Flamingo (Kong et al., 2024a) to generate ten captions for each audio sample and store the caption $c$ with the highest CLAP similarity $\cos(\text{CLAP}_{\text{audio}}(a), \text{CLAP}_{\text{text}}(c))$ to the audio $a$ (Wu et al., 2023). We discard the caption if the similarity is below 0.45, the optimal threshold according to AF-AudioSet (Kong et al., 2024b). In addition, there are challenges when applying this pipeline to larger-scale synthesis (beyond AudioSet), such as extremely long, homogeneous, or low-quality audio. To address these challenges, we caption each non-overlapping ten-second segment to obtain as many captions as possible. We then use keywords, e.g. "noisy", "low quality", or "unknown sounds", to detect low-quality audio. Finally, we also sub-sample long audio segments except for music and speech. With this strategy, we are able to generate 1.35M high-quality captions using audio from AudioCaps (Kim et al., 2019), AudioSet (Gemmeke et al., 2017), VGGSound (Chen et al., 2020), WavCaps (Mei et al., 2024), and Laion-630K (Wu et al., 2023). [3] We name our synthetic dataset **AF-Synthetic**.

Table 1 summarizes the comparison between AF-Synthetic and existing synthetic datasets. Our dataset is both large-scale (over 1M captions) and high-quality (CLAP $\geq 0.45$). We further apply our CLAP-similarity filtering to Sound-VECaps$_A$ (denoted as Sound-VECaps$_A$-0.45) and find that over 90% of the captions are rejected. Figure 1 displays the distributions of CLAP similarities. Our AF-Synthetic is over $8\times$ larger than Sound-VECaps$_A$-0.45 and AF-AudioSet, and has systematically higher CLAP similarities (about 3.8% absolute improvement on the median) than these two datasets. Table 9 in Appendix C.2 further shows that AF-Synthetic significantly improves text adherence of the audio compared to the baselines from human evaluations.

We then investigate the distributions of CLAP-similarity scores between our synthetic captions and AudioCaps and MusicCaps, two benchmarks we will use to evaluate our TTA. For each caption $c$ in AudioCaps or MusicCaps, we find its most similar caption $x$ from AF-Synthetic via the max-similarity $\text{max-sim}(X, c) = \max_{x \in X} \cos(\text{CLAP}_{\text{text}}(x), \text{CLAP}_{\text{text}}(c))$. We plot the distributions of max-sim in Table 2. We find AF-Synthetic has captions that are more similar to MusicCaps than AudioCaps, possibly due to caption lengths. We also find most max-sim scores are less than 0.9, indicating AF-Synthetic captions are quite different from these two datasets. We display some examples of most similar caption pairs in Appendix C.3. In summary:

---

[3]Our dataset has no overlap with MusicCaps (Agostinelli et al., 2023), which is also derived from AudioSet.

Table 1: Overview of our proposed AF-Synthetic dataset compared to existing synthetic captions datasets. AF-Synthetic improves the caption generation pipeline in AF-Audioset, and applies it to a variety of datasets, leading to a large-scale and high-quality synthetic dataset of captions. It is the first million-size synthetic captions dataset with strong audio-caption correlations (1.35M captions with CLAP similarity $\geq 0.45$). † After CLAP-similarity filtering.

| Dataset | Generation Model | Filtering Method | # Hours | # Captions |
|---|---|---|---|---|
| TangoPromptBank | Collected | None | 3.5K | 1.21M |
| Sound-VECaps$_A$ | CogVLM + EnClap | Removing visual-only data | 14.3K | 1.66M |
| Sound-VECaps$_A$-0.45$^\dagger$ | CogVLM + EnClap | CLAP $\geq 0.45$ | 448 | 161K |
| AutoCap | AutoCap | Removing music or speech | 8.7K | 761K |
| AF-AudioSet | Audio Flamingo | CLAP $\geq 0.45$ | 255 | 161K |
| AF-Synthetic (ours) | Audio Flamingo | CLAP $\geq 0.45$ and others | 3.6K | 1.35M |

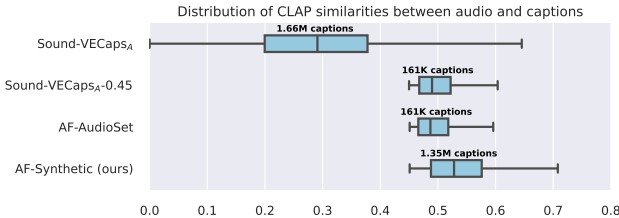

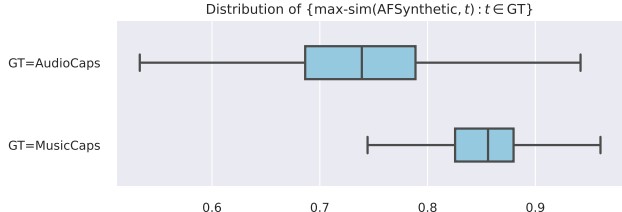

Figure 1: Distributions of CLAP similarities between audio $a$ and caption $c$, $\cos(\mathrm{CLAP}_{\mathrm{audio}}(a), \mathrm{CLAP}_{\mathrm{text}}(c))$, in existing datasets and our AF-Synthetic. Empirically, we consider a CLAP score of 0.4 as meaningful correlation, 0.45 stronger, and below 0.3 as weak. AF-Synthetic has >1M strongly correlated audio-caption pairs.

Figure 2: Distributions of max-similarities, $\mathrm{max\text{-}sim}(X, c) = \max_{x \in X} \cos(\mathrm{CLAP}_{\mathrm{text}}(x), \mathrm{CLAP}_{\mathrm{text}}(c))$, between AF-Synthetic and real datasets. Results indicate AF-Synthetic captions are quite different from AudioCaps and MusicCaps because most max-sim scores are below 0.9.

> AF-Synthetic is the first million-size synthetic caption dataset with strong audio correlations.

### 3.2. ETTA

Our TTA model, dubbed *Elucidated Text-To-Audio* (ETTA), is built upon the LDM (Rombach et al., 2022) paradigm and its application to audio generation. First, a variational autoencoder (VAE) (Kingma & Welling, 2014) is trained to compress waveform into a compact latent space. Once the VAE is trained, we freeze it and train a latent generative model in the VAE latent space. See Appendix B for mathematical details. We conduct our experiments based on the `stable-audio-tools` library,[4] which provides the most recent practices in building TTA models.

**ETTA-VAE** For training the VAE, we adopt a 44kHz stereo Audio-VAE with 156M parameters using the same default configuration used in `stable-audio-tools` with a latent frame rate of 21.5Hz. We refer to (Evans et al., 2024c) and Appendix B for details. The Audio-VAE is trained from scratch on our large-scale collection of publicly available datasets (see Table 12). In terms of quality, our Audio-VAE matches or exceeds Stable Audio Open, as

shown in Table 33 and Table 34 (Appendix G).[5]

**ETTA-LDM** Next, we train a text-conditional latent generative model for TTA synthesis. The latent model can be either a diffusion model (Ho et al., 2020; Song et al., 2021; Salimans & Ho, 2022) or a flow matching model (Lipman et al., 2022; Tong et al., 2023). We parameterize our model using the Diffusion Transformer (DiT) (Peebles & Xie, 2023) architecture based on Evans et al. (2024c) and Lan et al. (2024), with 24 layers, 24 heads, and a width of 1536 as the default choices. We condition our model on the outputs of the `T5-base` (Raffel et al., 2020) text encoder, which outputs embeddings for variable-length text. In our experiments, we also explore other common choices and combinations of different text encoders – including T5-based (Raffel et al., 2020; Chung et al., 2024) and CLAP models (Wu et al., 2023) – to study the effect of this component.

**ETTA-DiT** Finally, we provide several key improvements to the DiT implementation in Evans et al. (2024c), and call our implementation ETTA-DiT. Through experiments, we find that solely replacing their architecture with ETTA-DiT leads to improved training losses and evaluation results. Our

---

[4]https://github.com/Stability-AI/stable-audio-tools **commit id:** 7311840

[5]Since our dataset includes speech data, it is noticeably better in reconstructing speech signals.

improvements include:

*1)* Adaptive layer normalization (AdaLN): We switch from prepending to AdaLN timestep embedding and apply AdaLN. [6] Contrary to the baseline, we apply AdaLN to all inputs: self-attention, cross-attention, and feed-forward layer. The AdaLN parameters are initialized with scale $= 1$ and bias $= 0$ so that AdaLN does not modulate the feature at initialization. When applying AdaLN, we enforce `FP32`, use `torch.autocast` for numerical precision. Contrary to the baseline, we use a bias term for the linear layer and use unbounded gating (i.e. no `sigmoid`).

*2)* Final layers: Compared to the baseline, we initialize the final projection layer of DiT to output zeros. This matches the mean of the VAE latent distribution, and therefore leads to improved stability and convergence rate. We also use AdaLN in the final projection layer.

*3)* Other changes: we use the `tanh` approximation mode of the GELU activation (Hendrycks & Gimpel, 2016). We use rotary position embedding (RoPE) (Su et al., 2024) in the self-attention layer, with `rope_base` $= 16384$ to inject relative positional information. We also ensure that RoPE operates in `FP32`. We additionally apply dropout with $p_{\text{dropout}} = 0.1$ for all modules to enhance robustness in parameter estimation.

### 3.3. Training objective and Sampling

**Training**  For the diffusion model training objective, we use the $v$-prediction loss function (Salimans & Ho, 2022). For the flow matching training objective, we use the optimal transport conditional flow matching (OT-CFM) loss function (Lipman et al., 2022; Tong et al., 2023). We refer to Appendix B for details of these methods. Prior works also found sampling $t$ more often on intermediate steps leads to better results (Esser et al., 2024; Lan et al., 2024). We follow their approach and sample $t$ from a logit-normal distribution, in practice $t \sim \sigma(\mathcal{N}(0, 1))$, when training ETTA with OT-CFM.

**Sampling**  We consider Euler and $2^{\text{nd}}$-order Heun (Karras et al., 2022) methods for solving the ODE parameterized by ETTA. We conduct an extensive sweep over hyperparameters focusing on two major design choices: the number of function evaluations (NFE) and the classifier-free guidance (CFG) (Ho & Salimans, 2022) scale $w_{\text{cfg}}$. We draw Pareto curves across benchmark datasets and metrics to discover the optimal choice for ETTA. In addition, we also explore the effectiveness of a recently proposed guidance method, *autoguidance* (Karras et al., 2024), in TTA applications.

---

[6]In our preliminary study using `stable-audio-tools` with its vanilla implementation, switching from `prepending` to `AdaLN` resulted in worse results.

## 4. Experiments

Our experiments thoroughly evaluate our framework ETTA on benchmark datasets (AudioCaps and MusicCaps). We start with a systematic comparison to elucidate the design space of TTA in four major aspects: 1) training data, 2) training objectives, 3) architectural design and model sizes, and 4) sampling methods. Furthermore, we show ETTA's improved ability to generate creative audio following complex and imaginative captions, a task that is more challenging than current benchmarks. In our commitment to fully elucidate all aspects of our investigation, we also document the additional directions we explored, including numerous additional ablations (in Appendix D and E) and mixed or negative results (in Appendix F). We train all models using 8 A100 GPUs.

### 4.1. Training Data

We train ETTA on four different training datasets to assess TTA quality: AudioCaps (50K captions), AF-AudioSet (161K captions), TangoPromptBank (1.21M captions), and our AF-Synthetic (1.35M captions). We fix audio length to 10 seconds and sampling rate to 44.1kHz in all these datasets.

### 4.2. Training Objective and Sampling

**Audio VAE**  We train a 44.1kHz stereo Audio-VAE based on `stable-audio-tools` with our collection of unlabeled and public audio datasets (Table 12). We train the Audio-VAE using AdamW (Loshchilov, 2017) with a peak learning rate of $1.5 \times 10^{-4}$ with exponential decay for 2.8M steps, with a total batch size of 64 with 1.5 seconds per sample. We train with full precision (`FP32`) to make the waveform compression model as accurate as possible. The latent dimension is 64 and the frame rate is 21.5 Hz.

**Training Objective and Architecture**  We train ETTA-LDM with ETTA-DiT as the backbone. We use the `T5-base` text embedding with `max_length=512` truncation to accommodate longer captions in AF-Synthetic.[7] We train with both $v$-diffusion and OT-CFM objectives, where we additionally apply logit-normal $t$-sampling for OT-CFM (see Section 3.3). Our final model is trained for 1M steps using AdamW with a peak learning rate of $10^{-4}$ with exponential decay and total batch size of 128 with 10 seconds per sample. For ablation studies, we train each model for 250k steps unless otherwise stated. We use `BF16` mixed-precision training (Micikevicius et al., 2017) and `flash-attention 2` (Dao et al., 2022) to maximize training throughput.

---

[7]Our reproduction of Stable Audio Open using AF-Synthetic dataset also uses the same `max_length=512` for a fair comparison.

**Sampling** For diffusion models, following (Evans et al., 2024c) we use the `dpmpp-3m-sde` sampler [8] and CFG scale $w_{\text{cfg}} = 7$. For OT-CFM models, we compare between Euler and $2^{\text{nd}}$-order Heun samplers and draw Pareto curves for each method with respect to the number of function evaluations (NFEs) and CFG scale. After this extensive sweep, we choose Euler sampling with NFE = 100, $w_{\text{cfg}} = 3.5$ for main results, and $w_{\text{cfg}} = 1$ (no classifier-free guidance) for ablation studies unless otherwise stated.

### 4.3. Results

**Metrics** We use a collection of established objective metrics for systematic evaluation. 1) Fréchet distance (FD) measures the distributional gap between generated and ground truth audios using features extracted from an audio classifier. We consider PANNs (Kong et al., 2020) ($\text{FD}_P$) and OpenL3 (Cramer et al., 2019) ($\text{FD}_O$). [9] 2) Kullback–Leibler divergence (KL) is an instance-level metric that measures the difference between the posterior distributions of audio events for the ground truth and generated audio samples. This metric helps assess how close the generated audio aligns with the ground truth on the single-sample level. We report KL using PaSST (Koutini et al., 2022) ($\text{KL}_S$) and PANNs ($\text{KL}_P$). 3) Inception Score (IS) evaluates the diversity and specificity of the generated samples without requiring ground truth. IS is calculated from the entropy of instance posteriors and the entropy of marginal posteriors, where a higher score reflects both better diversity and sharper class predictions. We use PANNs for IS ($\text{IS}_P$). 4) Finally, CLAP score measures the cosine similarity between text and audio embeddings, which indicates the correlation between the generated sample and the given prompt. For extensive evaluation, we use two CLAP models: $\text{CL}_L$ for LAION's `630k-best` checkpoint (Wu et al., 2023) following Vyas et al. (2023), and $\text{CL}_M$ for MS-CLAP `2023` version (Elizalde et al., 2023). We also perform 5-scale subjective evaluation from mechanical turk following conventional metrics: 1) OVL: an overall quality of sample without seeing captions, and 2) REL: a relevance of the sample to the provided caption.

**Main Results** Tables 2 and 3 present our main objective results on AudioCaps and MusicCaps, respectively. Overall, ETTA shows significant improvements compared to Stable Audio Open (the base model) for both benchmarks with a single model. Compared to other works, ETTA shows competitive $\text{FD}_P$, $\text{FD}_O$, and KL scores. Notably, it shows exceptionally high $\text{IS}_P$ for both general sounds and music, demonstrating improved diversity and clarity of the gener-

---
[8] Implementation available in `https://github.com/crowsonkb/k-diffusion`

[9] OpenL3 is the latest model with better embedding quality, and $\text{FD}_O$ can measure up to 48kHz stereo quality (Evans et al., 2024c).

ated samples. Objective scores on MusicCaps are significantly better than previous models using public datasets and comparable to music specialist models (Li et al., 2024a;b; Fei et al., 2024) that use proprietary data. Since $\text{FD}_O$ can measure stereo audio, Stable Audio Open and ETTA are noticeably better than previous mono models. Both $\text{CL}_L$ and $\text{CL}_M$ show a preference towards ETTA, where our improvements on $\text{CL}_M$ is more salient. Subjective scores (OVL/REL) of ETTA are competitive with or outperforms baselines and are consistent with the objective evaluation.

We then fine-tune ETTA on the AudioCaps training set (FT-AC) for 50k and 100k additional steps. We find ETTA can quickly adapt to the target distribution via fine-tuning. Table 2 shows that ETTA keeps approximating the target distribution with better $\text{FD}_P$, which is close to Audiobox Sound (Vyas et al., 2023) trained on proprietary dataset. It is noteworthy that this also comes at a cost of shifting to the target distribution as evidenced by Table 3, where ETTA-FT-AC-100k starts to show noticeable degradation for music generation. In summary, our results show that:

> ETTA is the SOTA text-to-audio and text-to-music generation model using only publicly available data. It is also comparable to models trained with proprietary and/or licensed data.

**Design Improvements** Tables 4 summarize important design choices that lead to significant improvements. We use $\text{FD}_P$, $\text{KL}_S$, and $\text{CL}_M$ on MusicCaps as a summary (Full results in Tables 13 and 14, Appendix D). First, we reproduce Stable Audio Open using AF-Synthetic without other modification (+AF-Synthetic). Results show noticeable improvements from training data. Then, we switch the DiT implementation to ours (+ETTA-DiT). Results again show significant improvements. Next, we switch the training method from diffusion to OT-CFM (+OT-CFM) with conventional uniform timestep sampling ($t \sim \mathcal{U}(0,1)$). Empirically, although OT-CFM slightly degrades some metrics without CFG, we find OT-CFM is more stable to train, more consistent in quality especially with CFG, and more robust under fewer sampling steps in agreement with previous works. Finally, we adopt logit-normal $t$-sampling ($t \sim \sigma(\mathcal{N}(0,1))$) (Esser et al., 2024) and find it marginally improves $\text{FD}_P$. Therefore, we conclude:

> Our AF-Synthetic leads to the most significant improvements in ETTA. Our improved ETTA-DiT, the OT-CFM objective, and logit-normal $t$-sampling lead to further improvements.

**Scalability with Data** We assess the scalability of TTA models with respect to training data in Table 5 (Full results in Tables 21 and 22 in the Appendix E). First, AudioCaps lacks in quantity: ETTA trained solely on AudioCaps significaly underperforms on MusicCaps. TangoPromptBank

Table 2: Main results of ETTA compared to SOTA baselines on *AudioCaps*. FT-AC-$m$: fine-tuned on AudioCaps training set for $m$ iterations. ⋆ Best reported numbers. † Uses proprietary data.

| | $FD_P\downarrow$ | $FD_O\downarrow$ | $KL_S\downarrow$ | $KL_P\downarrow$ | $IS_P\uparrow$ | $CL_L\uparrow$ | $CL_M\uparrow$ | OVL↑ | REL↑ |
|---|---|---|---|---|---|---|---|---|---|
| Ground Truth | – | – | – | – | 13.49 | 0.62 | 0.38 | 3.43 ± 0.11 | 3.62 ± 0.10 |
| Audiobox (Vyas et al., 2023)⋆† | 10.14 | – | – | 1.19 | 11.90 | 0.70 | – | – | – |
| Audiobox Sound (Vyas et al., 2023)⋆† | 8.30 | – | – | 1.15 | 12.70 | 0.71 | – | – | – |
| Make-An-Audio (Huang et al., 2023c)⋆ | 18.32 | – | — | 1.61 | 7.29 | – | – | – | – |
| Make-An-Audio 2 (Huang et al., 2023a)⋆ | 11.75 | – | – | 1.32 | 11.16 | – | – | – | – |
| AudioLDM-L-Full (Liu et al., 2023b) | 23.31 | – | – | 1.59 | 8.13 | 0.43 | – | – | – |
| AudioLDM2 (Liu et al., 2024) | 26.44 | 156.64 | 1.81 | 1.79 | 8.14 | 0.50 | 0.36 | – | – |
| AudioLDM2-large (Liu et al., 2024)⋆ | 32.50 | 170.31 | 1.57 | 1.54 | 8.55 | 0.45 | – | – | – |
| AudioLDM2-large (Liu et al., 2024) | 26.18 | 158.05 | 1.68 | 1.64 | 8.55 | 0.53 | 0.37 | 3.00 ± 0.11 | 3.11 ± 0.10 |
| TANGO-Full-FT-AC (Ghosal et al., 2023)⋆ | 18.47 | – | 1.20 | 1.15 | 8.80 | 0.56 | – | – | – |
| TANGO-AF&AC-FT-AC (Kong et al., 2024b)⋆ | 17.19 | – | – | – | 11.04 | 0.53 | – | – | – |
| TANGO2 (Majumder et al., 2024)⋆ | – | – | – | 1.12 | 9.09 | – | – | 3.08 ± 0.10 | 3.66 ± 0.09 |
| GenAU-L (Haji-Ali et al., 2024)⋆ | 16.51 | – | – | – | 11.75 | – | – | – | – |
| Stable Audio Open (Evans et al., 2024c)⋆ | – | 78.24 | 2.14 | – | – | – | – | – | – |
| Stable Audio Open (Evans et al., 2024c) | 38.27 | 105.88 | 2.23 | 2.32 | 12.09 | 0.35 | 0.34 | 3.29 ± 0.11 | 3.15 ± 0.11 |
| *ETTA* | 13.12 | 80.13 | 1.22 | 1.42 | 14.36 | 0.54 | **0.43** | **3.43 ± 0.11** | 3.68 ± 0.10 |
| *ETTA*-FT-AC-50k | 11.13 | 65.35 | **1.12** | 1.26 | 15.05 | 0.59 | **0.43** | – | – |
| *ETTA*-FT-AC-100k | **10.10** | **61.79** | 1.13 | 1.24 | 14.29 | 0.60 | **0.43** | 3.26 ± 0.10 | **3.77 ± 0.10** |
| *ETTA* (AudioCaps only) | 12.21 | 71.84 | 1.19 | 1.30 | 10.07 | 0.58 | 0.40 | – | – |

Table 3: Main results of ETTA compared to SOTA baselines on *MusicCaps*. FT-AC-$m$: fine-tuned on AudioCaps training set for $m$ iterations. ⋆ Best reported numbers. † Uses proprietary or licensed data.

| | $FD_P\downarrow$ | $FD_O\downarrow$ | $KL_S\downarrow$ | $KL_P\downarrow$ | $IS_P\uparrow$ | $CL_L\uparrow$ | $CL_M\uparrow$ | OVL↑ | REL↑ |
|---|---|---|---|---|---|---|---|---|---|
| Ground Truth | – | – | – | – | 4.49 | 0.53 | 0.45 | 3.88 ± 0.10 | 3.90 ± 0.10 |
| Jen-1 (Li et al., 2024b)⋆† | – | – | – | 1.29 | – | – | – | – | – |
| QA-MDT (Li et al., 2024a)⋆† | – | – | – | 1.31 | 2.80 | – | – | – | – |
| FluxMusic (Fei et al., 2024)⋆† | – | – | – | 1.25 | 2.98 | – | – | – | – |
| MusicGen-medium (Copet et al., 2024)⋆ | – | – | 1.23 | 1.22 | – | – | – | – | – |
| AudioLDM-M (Liu et al., 2023b)⋆ | – | – | 1.29 | – | – | – | – | – | – |
| AudioLDM2 (Liu et al., 2024)⋆ | – | – | 1.20 | 1.20 | – | – | – | – | – |
| AudioLDM2 (Liu et al., 2024) | 21.39 | 198.45 | 1.19 | 1.57 | 2.48 | 0.45 | 0.45 | – | – |
| AudioLDM2-large (Liu et al., 2024) | 16.34 | 190.16 | 1.00 | 1.40 | 2.59 | 0.48 | 0.47 | 3.25 ± 0.10 | 3.15 ± 0.10 |
| TANGO-AF (Kong et al., 2024b) | 22.69 | 270.32 | 0.94 | 1.26 | 2.79 | **0.51** | 0.43 | 3.38 ± 0.09 | 3.31 ± 0.10 |
| Stable Audio Open (Evans et al., 2024c) | 36.42 | 127.20 | 1.32 | 1.56 | 2.93 | 0.48 | 0.49 | **3.92 ± 0.10** | 3.35 ± 0.11 |
| *ETTA* | **10.06** | 92.18 | **0.84** | **1.04** | **3.32** | **0.51** | **0.53** | 3.53 ± 0.10 | **3.57 ± 0.10** |
| *ETTA*-FT-AC-50k | 11.40 | 89.97 | 0.92 | 1.11 | 2.79 | 0.50 | **0.53** | – | – |
| *ETTA*-FT-AC-100k | 13.49 | **89.56** | 1.07 | 1.15 | 2.77 | 0.49 | 0.52 | 3.30 ± 0.10 | 3.44 ± 0.12 |

Table 4: Improvements by adding each of the major design choice of ETTA (evaluated on MusicCaps)

| Ablation | $FD_P\downarrow$ | $KL_S\downarrow$ | $CL_M\uparrow$ |
|---|---|---|---|
| Stable Audio Open | 39.96 | 1.81 | 0.41 |
| + AF-Synthetic | 26.22 | 1.57 | 0.43 |
| + ETTA-DiT | 20.48 | 1.38 | 0.45 |
| + OT-CFM, $t \sim \mathcal{U}(0,1)$ | 22.16 | 1.35 | 0.45 |
| + $t \sim \sigma(\mathcal{N}(0,1))$ | 21.59 | 1.41 | 0.45 |

Table 5: Ablation study on the results of ETTA trained on different datasets (evaluated on MusicCaps).

| Dataset (million captions) | $FD_P\downarrow$ | $KL_S\downarrow$ | $CL_M\uparrow$ |
|---|---|---|---|
| AudioCaps (0.05) | 76.14 | 3.20 | 0.27 |
| TangoPromptBank (1.21) | 24.72 | 1.73 | 0.38 |
| AF-AudioSet (0.16) | **21.40** | 1.45 | **0.44** |
| AF-Synthetic (1.35) | 21.59 | **1.41** | **0.44** |

is similar to AF-Synthetic in quantity:[10] while it scored

comparable $FD_P$, other metrics ($KL_S$ and $CL_M$) are much worse, suggesting that the quality of their music captions is not as good as AF-Synthetic. AF-AudioSet contains high-quality synthetic captions: it is competitive with AF-Synthetic, emphasizing the importance of data quality. We further evaluate on an out-of-distribution (OOD) dataset in Table 24 (Appendix E), and AF-Synthetic results are consistently better than AF-AudioSet. The results highlight that AF-Synthetic is a powerful dataset that is comprehensive in both quantity and quality. As such, we conclude:

> Both training data sizes and quality have positive effect on the results, where quality matters more.

**Scalability with Model Size** Table 6 provides the summary of scaling behavior of ETTA with respect to its model size. We explore different depths, widths, and the convolutional feed-forward layer kernel sizes ($k_{\text{convFF}}$) of ETTA-DiT. We use $w_{\text{cfg}} = 1$ to eliminate the effect of CFG.

As expected, most metrics show consistent improvements

---

[10]In practice, we used 2.33M audio-caption pairs for Tango-PromptBank due to repetitive captions for multiple 10-second

segments in a long audio.

Table 6: Ablation study on the results of ETTA with different depths, widths, and kernel sizes (evaluated on AudioCaps). ⋆ Our best model choice.

| Model | Size(B) | $\text{FD}_\text{P}\downarrow$ | $\text{KL}_\text{S}\downarrow$ | $\text{CL}_\text{M}\uparrow$ |
|---|---|---|---|---|
| depth = 4 | 0.38 | 36.46 | 2.15 | 0.30 |
| depth = 12 | 0.81 | 29.48 | 2.05 | 0.32 |
| depth = 24⋆ | 1.44 | 28.46 | 2.00 | 0.32 |
| depth = 36 | 2.08 | 27.08 | 1.95 | 0.32 |
| width = 384 | 0.28 | 35.97 | 2.14 | 0.30 |
| width = 768 | 0.52 | 31.03 | 2.04 | 0.32 |
| width = 1536⋆ | 1.44 | 28.46 | 2.00 | 0.32 |
| $k_\text{convFF} = 1$⋆ | 1.44 | 28.46 | 2.00 | 0.32 |
| $k_\text{convFF} = 3$ | 2.34 | 28.72 | 2.04 | 0.31 |

as we grow depth or width of ETTA-DiT. We find the 1.44B model with depth=24 and width=1536 leads to an optimal balance between model size and quality. On the other hand, increasing $k_\text{convFF}$ does not bring clear improvements, suggesting that allocating the model capacity to self-attention parameters is more important. See tables 17 and 18 (Appendix D) for extended results. In summary,

> In TTA tasks, increasing model size is helpful via increasing depth and width of DiT's self-attention block. However, increasing the kernel size of the convolutional feed-forward layer is not helpful.

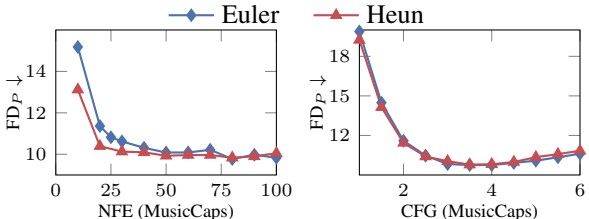

Figure 3: The effect of different sampling methods on the generation quality of ETTA MusicCaps. We investigate Euler and Heun solvers. NFE: number of function evaluations. CFG: classifier-free guidance scale.

**Choice of Sampler and its Impact on Metrics**  Figure 3 and Figure 5 (Appendix D) present a comprehensive analysis of the impact of sampler choices. The results reveal several key insights: 1) All metrics improve as the number of function evaluations (NFE) increases, as expected. 2) At lower NFE, the Heun sampler is noticeably better than Euler; as NFE increases, they converge to similar results. 3) FD behaves as a convex function with respect to the CFG scale, indicating that FD penalizes low diversity caused by CFG's over-emphasis on text condition and/or distortion caused by high CFG scales. 4) Metrics such as KL, IS, and CL (Figure 5) show continuous improvement with higher CFG scales, suggesting their preference for accuracy over diversity. Therefore, one should be cautious when selecting the CFG scale, as optimizing for these metrics alone may

lead to a trade-off between diversity and accuracy. Detailed results on the choices of sampler and NFE are provided in Table 19 (Appendix D). In summary:

> Heun's sampler is better than Euler at lower NFE. $w_\text{cfg} = 3.5$ provides the best overall metrics, and one should be cautious that a higher CFG scale potentially leads to lower diversity.

Table 7: Subjective Evaluation Result of Creative Audio Generation with 95% Confidence Interval.

| Model | AudioLDM2 | TANGO2 | Stable Audio Open | *ETTA* |
|---|---|---|---|---|
| OVL↑ | $3.95 \pm 0.05$ | $3.82 \pm 0.05$ | $3.94 \pm 0.05$ | $\mathbf{3.99} \pm 0.05$ |
| REL↑ | $3.79 \pm 0.06$ | $3.94 \pm 0.05$ | $3.95 \pm 0.05$ | $\mathbf{4.05} \pm 0.05$ |

**Creative Audio Generation**  We test ETTA's abilities to generate *creative* audio and music samples that do not exist in the real world, especially for complex and imaginative captions. We ask ChatGPT to generate hard captions that require blending and transformation of various sound elements towards creative audio. See Table 20 (Appendix D) for the imaginative captions. We generate 20 samples for each model and invite human listeners to measure 5-scale rating of OVL and REL. Table 7 shows that ETTA significantly improves its ability to follow the complex captions as measured by the REL score ($p < 0.05$ from Wilcoxon signed-rank test). We strongly encourage the readers to listen to the audio samples in the demo page (Appendix A). Therefore, we claim:

> ETTA shows an improved ability to generate audio that follows complex and imaginative captions.

## 5. Conclusion

In this paper, we setup a large-scale empirical experiment to comprehensively understand the design space of modern text-to-audio models. We provide insights on data scaling, architectural design, model scaling, training methods, and inference strategies. Based on our findings, we present ETTA, a state-of-the-art text-to-audio model that results from large-scale and high-quality synthetic captions, a better DiT implementation, and a better VAE.

**Future work**  While this work aims to elucidate the design space of TTA with large-scale experiments, there are still several unexplored problems we plan to study in our future work. (1) We plan to improve data augmentation with caption rephrasing and audio re-mixing (Melechovsky et al., 2023; Liu et al., 2023b; 2024; Huang et al., 2023c;a) and systematically study the effect of data augmentation. (2) We plan to investigate better evaluation methods and benchmarks for text-to-audio generation that could reflect both the accuracy and diversity of TTA models in a way that corresponds with human perception.

## Impact Statement

This paper aims to contribute to the advancement of generative modeling of audio by introducing a method that enhances the quality corresponding better to the input text prompt. The approach has broad applicability across industries, such as media production and music composition. However, responsible usage is crucial to ensure adherence to copyright regulations in specific contexts.

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

## A. Links

The link to our demo page is:
https://research.nvidia.com/labs/adlr/
ETTA/

The link to our code repository is:
https://github.com/NVIDIA/
elucidated-text-to-audio

## B. Mathematical Background

Let $p_{\text{data}}$ be the data distribution and $X \sim p_{\text{data}}$ be $N$ i.i.d. training samples drawn from the data distribution. In (unconditional) audio synthesis, we assume $p_{\text{data}}$ is on $[-1, 1]^L$ where $L = 441000$ is a fixed length for 10 seconds of audio at 44.1kHz sampling rate. A generative model on $X$ aims to model $p_\theta(x) \approx p_{\text{data}}(x)$ and draw samples from it. In text-to-audio synthesis, each sample $x = (a, c)$ is composed of an audio $a \in [-1, 1]^L$ and a corresponding caption $c$ in the natural language space. In this case, we aim to model $p_\theta(a|c)$ and draw samples conditioned on a given caption $c$. For conciseness, we introduce all the mathematical background in the unconditional setting, and these can be translated into the conditional setting by conditioning all distributions on $c$.

### B.1. Variational Auto Encoders

Variational auto encoders (VAEs) (Kingma & Welling, 2014) include an encoder $E$ and a decoder $D$. $E$ aims to encode a sample $x$ into a lower-dimensional space, and $D$ aims to reconstruct $E(x)$ to the original space with minimal information loss. The training loss is

$$L_{\text{VAE}} = \mathbb{E}_{x \sim X}[\mathcal{R}(D(E(x)), x) + \text{KL}(q_E(z|x) \| \mathcal{N}(0, I))],$$

where $\mathcal{R}$ is a reconstruction loss that measures the distance between the original sample $x$ and the reconstructed sample $D(E(x))$. $q_E(z|x)$ is the approximate posterior distribution of the latent variable $z$ given $x$ using $E$, and the KL divergence loss measures how close the posterior distribution is to the prior $\mathcal{N}(0, I)$.

Stable Audio Open's VAE (Evans et al., 2024c) is trained with a combination of below losses:

1. A stereo sum and difference multi-resolution STFT loss (Steinmetz & Reiss, 2020; Steinmetz et al., 2021) that computes distances in the spectrogram space with different resolutions:

$$L_{\text{MRSTFT}}(x, \hat{x}) = \sum_{i=1}^{m} \left( \frac{\|\text{stft}_i(x) - \text{stft}_i(\hat{x})\|_F}{\|\text{stft}_i(x)\|_F} \right.$$
$$\left. + \frac{1}{T} \left\| \log \frac{\text{stft}_i(x)}{\text{stft}_i(\hat{x})} \right\|_1 \right), \quad (1)$$

$$L_{\text{StereoMRSTFT}}(x, \hat{x}) = L_{\text{MRSTFT}}(x_{\text{sum}}, \hat{x}_{\text{sum}})$$
$$+ L_{\text{MRSTFT}}(x_{\text{diff}}, \hat{x}_{\text{diff}}), \quad (2)$$

where $T$ is the number of STFT frames, each $\text{stft}_i$ is the STFT transformation with resolution $i$, and

$$x_{\text{sum}} = x_{\text{left}} + x_{\text{right}}, \quad x_{\text{diff}} = x_{\text{left}} - x_{\text{right}}. \quad (3)$$

2. An adversarial hinge loss and feature matching loss from Encodec (Défossez et al., 2023):

$$L_{\text{adv}}(\hat{x}, x) = \sum_{k=1}^{K} \Big( \max(0, 1 - D_k(x))$$
$$+ \max(0, 1 + D_k(\hat{x})) \Big), \quad (4)$$

$$L_{\text{feat}}(x, \hat{x}) = \frac{1}{KL} \sum_{k=1}^{K} \sum_{l=1}^{L} \frac{\|D_k^l(x) - D_k^l(\hat{x})\|_1}{\text{mean}(\|D_k^l(x)\|_1)}, \quad (5)$$

where $D_k^l$ is the $l$-th layer of the $k$-th discriminator $D_k$.

3. The KL divergence loss:

$$\text{KL}(q_E(z|x) \| \mathcal{N}(0, I)). \quad (6)$$

The VAE is trained using randomly chunked unlabeled audio data without captions.

### B.2. Diffusion Models

Diffusion models (Ho et al., 2020; Song et al., 2021) include two processes:

1. A fixed Markov chain diffusion process

$$dx = \mathbf{f}(x, t)dt + g(t)d\mathbf{w},$$

where $x$ represents data, $t \in [0, 1]$ represents time, $\mathbf{f}$ is the drift term, $g$ is the diffusion term, and $d\mathbf{w}$ is the standard Brownian motion.

2. A learned Markov chain reverse process

$$dx = [\mathbf{f}(x, t) - g(t)^2 \nabla_x \log p_t(x)]dt + g(t)d\bar{\mathbf{w}},$$

where $d\bar{\mathbf{w}}$ is the reverse Brownian motion.

A neural network $s_\theta(x, t)$ is used to substitute the score function $\nabla_x \log p_t(x)$ and therefore trained to approximate the true score function $\nabla_x \log q(x|x_0)$ at time $t$, leading to training objective

$$\mathbb{E}_{t \sim \mathcal{U}(0,1), x_0 \sim p_{\text{data}}, x_t \sim q(x_t|x_0)} \| s_\theta(x_t, t) - \nabla_{x_t} \log q(x_t|x_0) \|^2,$$

where we could write $x_t$ in terms of noise $\epsilon_t \sim \mathcal{N}(0, I)$ : $x_t = \sqrt{\alpha_t} x_0 + \sqrt{1 - \alpha_t} \epsilon_t$ for a pre-defined schedule $\alpha_t$,

Table 8: Detailed breakdown of our proposed AF-Synthetic dataset compared to existing datasets.

| Dataset | Total Hours | Number of captions | | | | | |
|---|---|---|---|---|---|---|---|
| | | Total | AudioCaps | AudioSet | Laion-630K | WavCaps | VGGSound |
| TangoPromptBank | 3.5K | 1.21M | 45K | 108K | - | 1.05M | - |
| Sound-VECaps$_A$ | 14.3K | 1.66M | - | 1.66M | - | - | - |
| AutoCap | 8.7K | 761K | - | 339K | 295K | - | 127K |
| AF-AudioSet | 255 | 161K | - | 161K | - | - | - |
| AF-Synthetic | 3.6K | 1.35M | 33K | 165K | 282K | 783K | 92K |

and $\nabla_{x_t} \log q(x_t|x_0) = -\epsilon_t/\sqrt{1-\alpha_t}$. For this reason, the standard loss function is called the $\epsilon$-prediction.

One can predict other quantities to train diffusion models as well. One example is the $x$-diffusion, where we train a network to predict $\hat{x}_t = (x_t - \sqrt{1-\alpha_t}\epsilon_t)/\sqrt{\alpha_t}$. Another example is the $v$-diffusion (Salimans & Ho, 2022), where the network predicts $\hat{v}_t = \sqrt{\alpha_t}\epsilon_t - \sqrt{1-\alpha_t}x_0$.

### B.3. Optimal Transport Conditional Flow Matching

Optimal Transport Conditional Flow Matching (OT-CFM) (Lipman et al., 2022; Tong et al., 2023) is an alternative method to train diffusion models via flow matching. Instead of predicting $\epsilon$ it directly predicts the vector field $\mathbf{f}(x,t) - g(t)^2 \nabla_x \log p_t(x)$, leading to the following loss function:

$$L_{\text{OTCFM}} = \mathbb{E}_{t\sim U(0,1), x_0\sim p_{\text{data}}, x_t\sim q(x_t|x_0)}$$
$$\left\| v_\theta(x_t,t) - \left(\mathbf{f}(x_t,t) - g(t)^2 \nabla_{x_t} \log q(x_t|x_0)\right) \right\|^2.$$

### B.4. Latent Diffusion Models

Latent diffusion models (LDMs) (Rombach et al., 2022; Liu et al., 2023b) combine VAE with diffusion models, training the diffusion models within the latent space of the VAE. In this approach, the VAE's latent variable $z$ serves as the target for generation. Rather than directly modeling $p_{\text{data}}$, LDMs model the pushforward distribution $E_{\#}p_{\text{data}}$, utilizing the frozen encoder and decoder from the VAE to transition between the original data space and the latent space.

### B.5. Classifier-Free Guidance

Classifier-Free Guidance (CFG) (Ho & Salimans, 2022) adjusts the balance between diversity and quality in generative models by over-emphasizing conditioning. The model is trained both conditionally and unconditionally by randomly replacing the condition $c$ with a null embedding $\emptyset$. During sampling, the guided output is given by:

$$v_\theta(x_t,t|c) = v_\theta(x_t,t) + w_{\text{cfg}} \cdot (v_\theta(x_t,t|c) - v_\theta(x_t,t)),$$

where $w_{\text{cfg}}$ is a guidance scale. $w_{\text{cfg}} = 1$ disables guidance and $w_{\text{cfg}} > 1$ amplifies the conditioning.

## C. Dataset Details

### C.1. AF-Synthetic Details

Table 8 provides a detailed breakdown of sources of data from which each audio-caption dataset is built. Compared to previous datasets, AF-Synthetic include diverse data source to construct synthetic captions, which enables strong generalization to numerous audio types when training TTA model.

We use Laion-CLAP `630k-audioset-fusion-best` checkpoint to compute CLAP similarity (Wu et al., 2023). We use the following keywords to filter low-quality audio samples:

```
ambiguous, artifact, background noise, broken up,
buzzing, choppy, clipping, compromised, crackling,
deficient, distant, distorted, dropout, echo,
faint, faulty, feedback, flawed, fluctuating, fuzzy,
garbled, gibberish, glitch, hissing, imprecise,
inadequate, inaudible, incoherent, indistinct,
inferior, insufficient, interference, irregular,
irrelevant, lacking, low quality, low volume,
low-quality, mediocre, misheard, misinterpretation,
muffled, murmur, noise, noisy, off-mic, overlapping
speech, overmodulated, poor, popping, reverberation,
scrambled, second-rate, sibilance, skipped, skipping,
static, suboptimal, substandard, uncertain, unclear,
undermodulated, unintelligible, unknown sounds,
unreliable, unsatisfactory, unspecific, vague.
```

### C.2. Subjective Evaluation of AF-Synthetic Captions

Table 9 shows the 5-scale REL scores of AF-Synthetic compared to the original baseline captions for three subsets: 1) AudioSet, where the baseline captions are drawn from Laion-630k, 2) FreeSound subset of Laion-630k along with the original captions, and 3) WavCaps, where the captions are from LLM rephrasing of the original metadata. We randomly sampled 100 audio-caption pairs for each subset and used the same human evaluation protocol to measure the REL scores. For all subsets we consider, AF-Synthetic gives significant improvements in text adherence by human raters, consistent with the objective results using CLAP similarity.

## C.3. Most Similar AF-Synthetic Captions to AudioCaps and MusicCaps

In Table 10 and Table 11 we show some captions from AudioCaps or MusicCaps and their most similar captions from AF-Synthetic. These examples, together with Figure 2, demonstrate that AF-Synthetic captions are quite different from these two datasets, which further proves the generalization ability of our ETTA that is only trained on AF-Synthetic.

Table 9: REL Scores for AF-Synthetic Caption Quality with 95% Confidence Interval.

| REL↑ | AudioSet | FreeSound | WavCaps |
|---|---|---|---|
| Baseline | $3.82 \pm 0.11$ | $3.68 \pm 0.13$ | $3.92 \pm 0.11$ |
| AF-Synthetic | $\mathbf{4.04} \pm 0.10$ | $\mathbf{3.83} \pm 0.11$ | $\mathbf{4.02} \pm 0.10$ |

## C.4. Training Data for ETTA-VAE

Table 12 shows the training data of our ETTA-VAE.

# D. Additional Ablation Study on ETTA-DiT

Table 13 and 14 show an extended ablation study of our architectural design from ETTA-DiT. Figure 4 shows training loss comparisons to further justify the main design choices of ETTA. Tables 15 and 16 discuss additional setups we explored: setting a RoPE frequency base, the use of dropout. Tables 17 and 18 provide extended study regarding model capacity.

**Training loss comparison**     Figure 4 displays training loss plots over several configurations to illustrate the rationale behind each of the design choices of ETTA. The result shows that: 1) Stable Audio-DiT plateaus around 300K steps and starts to diverge early. ETTA-DiT continues to improve its quality with better loss for the same training steps. This shows clear benefits of the ETTA-DiT architecture. 2) For prolonged training (e.g. over 500K steps), v-diffusion starts to be unstable, whereas OT-CFM provides better stability up to 1M steps. This shows practical advantages of OT-CFM over v-diffusion and is a motivation to use it for ETTA. 3) AF-AudioSet quickly diverges around 250K steps and is unable to continue its training, whereas AF-Synthetic provides better convergence with continued improvements up to 1M steps, meaning that AF-Synthetic helps ETTA converge better from its scale.

**RoPE frequency base**     We decide to use `rope_base=16384` which can be considered as significantly "longer" than the length ETTA would usually be exposed to (up to 512 for text token embedding, and 215 for the VAE latent window). This design is inspired by recent trends in LLM where applying longer `rope_base` during training helps improving extrapolation to longer sequence generation. Considering usual I/O length of ETTA, we also tried using shorter `rope_base=512`. We find that the early training loss is slightly better but the difference in objective metrics is small, mostly within an expected margin of error. While the shorter `rope_base` may have been sufficient, our final model uses the longer one towards scalability to longer text and audio window beyond what we have explored in this work.

> Different RoPE frequency base does not affect the results significantly. However, we conjecture longer value can help for models with longer window.

**Dropout**     Although turning off dropout $p_{\text{dropout}} = 0.0$ yields slightly better benchmark scores (FD scores and $\text{KL}_S$, for example) measured at 250k training steps, we decide to use $p_{\text{dropout}} = 0.1$ for the final model where we speculate that it may provide improved generalization and enhance robustness in parameter estimation, leading to a more robust model in real-world captions beyond benchmark datasets. We do not draw a conclusion that turning off dropout is better or worse in this work, and it remains to be seen if it

Table 10: Examples of captions from AudioCaps and their most similar caption from AF-Synthetic.

| AudioCaps caption | Most similar AF-Synthetic caption |
|---|---|
| An airplane engine running. | The audio primarily features the continuous roar of an aircraft engine, with a high-pitched whoosh, swoosh, or swish sound also present. |
| Multiple cars are racing, speeding and roaring in the distance. | The audio features the distinct sounds of a race car and other racing vehicles. The race car engine is the dominant sound throughout the audio, while the other racing vehicles can be heard intermittently. |
| A consistent, loud mechanical motor. | The audio features an aircraft engine, which produces a loud, continuous, mechanical sound. The wind sound is also audible throughout the audio. |
| A small tool motor buzzes and an adult male speaks. | The audio features a man speaking intermittently, with the sound of an electric shaver running throughout. There are also instances of a high-pitched beeping sound. |
| A mid-size motor vehicle engine is idling. | The audio primarily consists of the sound of a large truck engine idling, with occasional engine revving sounds. There is also a high frequency, random-frequency content present throughout the audio. |
| Insect noises with people talking. | The audio features a child speaking, with the sound of insects and background noise throughout. There's also a brief sound of a buzzing, repetitive cricket. |
| A very short spray and then silence after that. | The audio contains the sound of a spark and a hiss, which are often heard when a spark is created in a gas or a fluid. |
| Multiple dogs bark, people speak. | The audio features a dog barking and yipping, along with the sound of a television playing in the background. There's also a conversation happening, with a woman speaking at certain intervals. Additionally, there are instances of a human voice and laughter. |

Table 11: Examples of captions from AudioCaps and their most similar caption from AF-Synthetic.

| MusicCaps caption | Most similar AF-Synthetic caption |
|---|---|
| The low quality recording features a ballad song that contains sustained strings, mellow piano melody and soft female vocal singing over it. It sounds sad and soulful, like something you would hear at Sunday services. | The audio features a calming piano melody and soft vocals. |
| A male voice is singing a melody with changing tempos while snipping his fingers rhythmically. The recording sounds like it has been recorded in an empty room. This song may be playing, practicing snipping and singing along. | The audio features a male voice, which is singing a catchy melody with a folk style. |
| This song contains digital drums playing a simple groove along with two guitars. ne strumming chords along with the snare the other one playing a melody on top. An e-bass is playing the footnote while a piano is playing a major and minor chord progression. A trumpet is playing a loud melody alongside the guitar. All the instruments sound flat and are being played by a keyboard. There are little bongo hits in the background panned to the left side of the speakers. Apart from the music you can hear eating sounds and a stomach rumbling. This song may be playing for an advertisement. | The audio features a synth, drums, and a guitar. The synth is playing a repetitive melody, the drums are playing a beat, and the guitar is strumming chords. |
| This clip is three tracks playing consecutively. The first one is an electric guitar lead harmony with a groovy bass line, followed by white noise and then a female vocalisation to a vivacious melody with a keyboard harmony, slick drumming, funky bass lines and male backup. The three songs are unrelated and unsynced. | The audio contains a distorted rock song, playing on top of acoustic drums. There are also sounds of a crowd and clapping, which contribute to the overall energetic and lively feel of the music. |
| A male singer sings this groovy melody. The song is a techno dance song with a groovy bass line, strong drumming rhythm and a keyboard accompaniment. The song is so groovy and serves as a dance track for the dancing children. The audio quality is very poor with high gains and hissing noise. | The audio features a strong bass and electronic drum beats, which are characteristic of this genre. There's also the sound of a female voice singing, which adds a unique element to the overall sound. |
| Someone is playing a high pitched melody on a steel drum. The file is of poor audio-quality. | The audio features a steelpan being played to music. |
| Low fidelity audio from a live performance featuring a solo direct input acoustic guitar strumming airy, suspended open chords. Also present are occasional ambient sounds, perhaps papers being shuffled. | The audio features the sustained, mellow strumming of a nylon string guitar in free time. There are also high pitched, thin strings being plucked. |
| The instrumental music features an ensemble that resembles the orchestra. The melody is being played by a brass section while strings provide harmonic accompaniment. At the end of the music excerpt one can hear a double bass playing a long note and then a percussive noise. | The audio features a variety of strings and brass instruments playing a fast melody. |

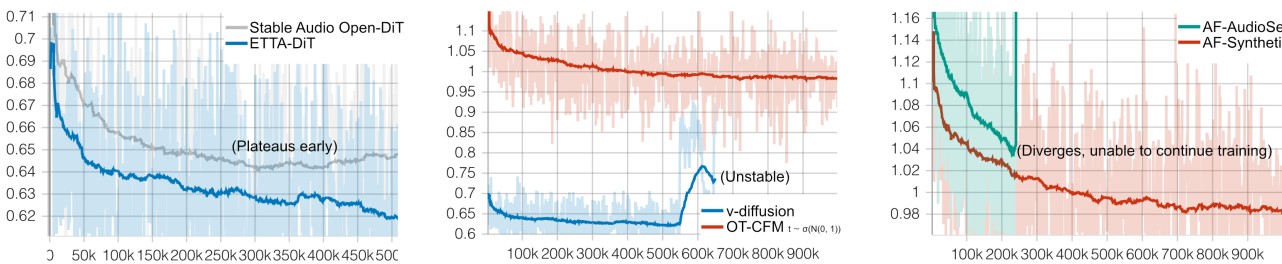

Figure 4: Training loss comparison across three setups: (left) Stable Audio Open-DiT vs. ETTA-DiT, using the same v-diffusion objective and AF-Synthetic training dataset. (center) v-diffusion vs. OT-CFM, using the same ETTA-DiT architecture and AF-Synthetic training dataset. (right) AF-AudioSet vs. AF-Synthetic, using the same OT-CFM training objective and ETTA-DiT architecture.

would help or not as we scale data and model further.

Dropout does not affect the overall results significantly. We speculate that adding dropout could enhance robustness in parameter estimation as we scale the TTA models.

**Scalability with Model Size while CFG turned on** Tables 17 and 18 show additional result on the model size scaling experiment using $w_{cfg} = 1$ or $w_{cfg} = 3$. Compared to $w_{cfg} = 1$, the difference of metrics between model of different sizes is smaller. This suggests that while the quality

Table 12: Datasets used for training ETTA-VAE.

| Dataset | URL |
|---|---|
| HiFi-TTS | https://www.openslr.org/109/ |
| MSP-PODCAST-Publish-1.9 | https://ecs.utdallas.edu/research/researchlabs/msp-lab/MSP-Podcast.html |
| SIWIS | https://datashare.ed.ac.uk/handle/10283/2353 |
| Spanish-HQ | https://openslr.org/72/ |
| TTS-Portuguese-Corpus | https://github.com/Edresson/TTS-Portuguese-Corpus |
| VCTK | https://datashare.ed.ac.uk/handle/10283/3443 |
| css10 | https://github.com/Kyubyong/css10 |
| indic-languages-tts-iiit-h | http://festvox.org/databases/iiit_voices/ |
| l2arctic | https://psi.engr.tamu.edu/l2-arctic-corpus/ |
| CREMA-D | https://github.com/CheyneyComputerScience/CREMA-D |
| emov-db | https://github.com/numediart/EmoV-DB |
| jl-corpus | https://github.com/tli725/JL-Corpus |
| ravdess | https://www.kaggle.com/datasets/uwrfkaggler/ravdess-emotional-speech-audio |
| tess | https://www.kaggle.com/datasets/ejlok1/toronto-emotional-speech-set-tess |
| AudioSet | https://research.google.com/audioset/download.html |
| Laion630k-audio (2022) | https://github.com/LAION-AI/audio-dataset |
| Clotho-AQA | https://zenodo.org/records/6473207 |
| Clotho-v2 | https://github.com/audio-captioning/clotho-dataset/tree/master |
| CochlScene | https://github.com/cochlearai/cochlscene |
| DCASE17Task4 | https://dcase.community/challenge2017/task-large-scale-sound-event-detection-results |
| ESC-50 | https://github.com/karolpiczak/ESC-50 |
| FMA | https://github.com/mdeff/fma |
| FSD50k | https://zenodo.org/records/4060432 |
| GTZAN | https://www.tensorflow.org/datasets/catalog/gtzan |
| IEMOCAP | http://sail.usc.edu/iemocap/ |
| MACS | https://zenodo.org/records/5114771 |
| MELD | https://github.com/declare-lab/MELD |
| MU-LLAMA | https://github.com/shansongliu/MU-LLaMA?tab=readme-ov-file |
| MagnaTagATune | https://mirg.city.ac.uk/codeapps/the-magnatagatune-dataset |
| Medley-solos-DB | https://zenodo.org/records/3464194 |
| Music-AVQA | https://gewu-lab.github.io/MUSIC-AVQA/ |
| MusicNet | https://www.kaggle.com/datasets/imsparsh/musicnet-dataset |
| NSynth | https://magenta.tensorflow.org/datasets/nsynth |
| NonSpeech7k | https://zenodo.org/records/6967442 |
| OMGEmotion | https://www2.informatik.uni-hamburg.de/wtm/OMG-EmotionChallenge/ |
| OpenAQA | https://github.com/YuanGongND/ltu?tab=readme-ov-file#openaqa-ltu-and-openasqa-ltu-as-dataset |
| SONYC-UST | https://zenodo.org/records/3966543 |
| SoundDescs | https://github.com/akoepke/audio-retrieval-benchmark |
| UrbanSound8K | https://urbansounddataset.weebly.com/urbansound8k.html |
| VocalSound | https://github.com/YuanGongND/vocalsound |
| WavText5K | https://github.com/microsoft/WavText5K |
| AudioCaps | https://github.com/cdjkim/audiocaps |
| chime-home | https://code.soundsoftware.ac.uk/projects/chime-home-dataset-annotation-and-baseline-evaluation-code |
| common-accent | https://huggingface.co/datasets/DTU54DL/common-accent |
| maestro-v3 | https://magenta.tensorflow.org/datasets/maestro |
| mtg-jamendo | https://github.com/MTG/mtg-jamendo-dataset |
| MUSDB-HQ | https://zenodo.org/records/3338373 |

Table 13: Improvements by adding each of the major design choices of ETTA (evaluated on AudioCaps).

| Ablation | $FD_P\downarrow$ | $FD_O\downarrow$ | $KL_S\downarrow$ | $KL_P\downarrow$ | $IS_P\uparrow$ | $CL_L\uparrow$ | $CL_M\uparrow$ |
|---|---|---|---|---|---|---|---|
| Stable Audio Open | 47.10 | 127.82 | 3.14 | 3.13 | 6.81 | 0.18 | 0.24 |
| + AF-Synthetic | 37.40 | 125.33 | 2.45 | 2.69 | 5.37 | 0.28 | 0.29 |
| + ETTA-DiT | 28.20 | 92.31 | 2.07 | 2.19 | 6.04 | 0.37 | 0.33 |
| + OT-CFM, $t \sim \mathcal{U}(0,1)$ | 30.39 | 89.44 | 2.03 | 2.26 | 5.48 | 0.37 | 0.31 |
| + $t \sim \sigma(\mathcal{N}(0,1))$ | 28.46 | 89.60 | 1.99 | 2.21 | 5.64 | 0.37 | 0.32 |

Table 14: Improvements by adding each of the major design choices of ETTA (evaluated on MusicCaps).

| Ablation | $FD_P\downarrow$ | $FD_O\downarrow$ | $KL_S\downarrow$ | $KL_P\downarrow$ | $IS_P\uparrow$ | $CL_L\uparrow$ | $CL_M\uparrow$ |
|---|---|---|---|---|---|---|---|
| Stable Audio Open | 39.96 | 119.54 | 1.81 | 2.11 | 3.19 | 0.34 | 0.41 |
| + AF-Synthetic | 26.22 | 127.90 | 1.57 | 1.73 | 2.37 | 0.39 | 0.43 |
| + ETTA-DiT | 20.48 | 100.53 | 1.38 | 1.50 | 2.21 | 0.42 | 0.45 |
| + OT-CFM, $t \sim \mathcal{U}(0,1)$ | 22.16 | 98.84 | 1.35 | 1.49 | 2.10 | 0.42 | 0.45 |
| + $t \sim \sigma(\mathcal{N}(0,1))$ | 21.59 | 92.30 | 1.41 | 1.51 | 2.20 | 0.41 | 0.45 |

of model grows with its total size, small models can also generate high-quality samples with CFG at a cost of having potentially lower diversity.

> Classifier-free guidance helps smaller models to be closer to large models in objective metrics.

**Full results on the sampling methods** Table 19 and Figure 5 show the full results on the effect of sampling methods,

Table 15: Ablation study on the effect of other architectural designs of ETTA on generation quality (evaluated on AudioCaps).

| Ablation | $FD_P\downarrow$ | $FD_O\downarrow$ | $KL_S\downarrow$ | $KL_P\downarrow$ | $IS_P\uparrow$ | $CL_L\uparrow$ | $CL_M\uparrow$ |
|---|---|---|---|---|---|---|---|
| ETTA | 13.01 | 81.23 | 1.29 | 1.50 | 12.42 | 0.52 | 0.41 |
| ETTA + `rope_base` $= 512$ | 12.64 | 79.49 | 1.32 | 1.51 | 12.45 | 0.52 | 0.41 |
| ETTA + $p_{\mathrm{dropout}} = 0.0$ | 13.04 | 76.30 | 1.28 | 1.50 | 12.27 | 0.53 | 0.41 |

Table 16: Ablation study on the effect of other architectural designs of ETTA on generation quality (evaluated on MusicCaps).

| Ablation | $FD_P\downarrow$ | $FD_O\downarrow$ | $KL_S\downarrow$ | $KL_P\downarrow$ | $IS_P\uparrow$ | $CL_L\uparrow$ | $CL_M\uparrow$ |
|---|---|---|---|---|---|---|---|
| ETTA | 12.15 | 96.46 | 0.88 | 1.08 | 2.93 | 0.51 | 0.52 |
| ETTA + `rope_base` $= 512$ | 12.11 | 95.04 | 0.84 | 1.08 | 2.94 | 0.51 | 0.52 |
| ETTA + $p_{\mathrm{dropout}} = 0.0$ | 11.75 | 88.74 | 0.75 | 1.10 | 2.97 | 0.51 | 0.51 |

Table 17: Ablation study on the results of ETTA with different depths, widths, and kernel sizes (evaluated on AudioCaps). The classifier-free guidance $w_{\mathrm{cfg}} = 1$. $\star$ Our best model choice.

| Model | Size(B) | $FD_P\downarrow$ | $FD_O\downarrow$ | $KL_S\downarrow$ | $KL_P\downarrow$ | $IS_P\uparrow$ | $CL_L\uparrow$ | $CL_M\uparrow$ |
|---|---|---|---|---|---|---|---|---|
| `depth` $= 4$ | 0.38 | 36.46 | 103.35 | 2.15 | 2.39 | 4.88 | 0.33 | 0.30 |
| `depth` $= 12$ | 0.81 | 29.48 | 93.13 | 2.05 | 2.28 | 5.73 | 0.36 | 0.32 |
| `depth` $= 24^\star$ | 1.44 | 28.46 | 89.61 | 2.00 | 2.22 | 5.65 | 0.37 | 0.32 |
| `depth` $= 36$ | 2.08 | 27.08 | 82.60 | 1.95 | 2.18 | 5.87 | 0.38 | 0.32 |
| `width` $= 384$ | 0.28 | 35.97 | 100.58 | 2.14 | 2.43 | 4.99 | 0.33 | 0.30 |
| `width` $= 768$ | 0.52 | 31.03 | 93.74 | 2.04 | 2.29 | 5.49 | 0.36 | 0.32 |
| `width` $= 1536^\star$ | 1.44 | 28.46 | 89.61 | 2.00 | 2.22 | 5.65 | 0.37 | 0.32 |
| $k_{\mathrm{convFF}} = 1^\star$ | 1.44 | 28.46 | 89.61 | 2.00 | 2.22 | 5.65 | 0.37 | 0.32 |
| $k_{\mathrm{convFF}} = 3$ | 2.35 | 28.72 | 82.49 | 2.04 | 2.28 | 5.84 | 0.36 | 0.31 |

Table 18: Ablation study on the results of ETTA with different depths, widths, and kernel sizes (evaluated on AudioCaps). The classifier-free guidance $w_{\mathrm{cfg}} = 3$. $\star$ Our best model choice.

| Model | Size(B) | $FD_P\downarrow$ | $FD_O\downarrow$ | $KL_S\downarrow$ | $KL_P\downarrow$ | $IS_P\uparrow$ | $CL_L\uparrow$ | $CL_M\uparrow$ |
|---|---|---|---|---|---|---|---|---|
| `depth` $= 4$ | 0.38 | 15.81 | 81.71 | 1.41 | 1.55 | 11.35 | 0.50 | 0.40 |
| `depth` $= 12$ | 0.81 | 13.88 | 83.62 | 1.36 | 1.54 | 12.52 | 0.52 | 0.41 |
| `depth` $= 24^\star$ | 1.44 | 13.01 | 81.23 | 1.29 | 1.50 | 12.42 | 0.52 | 0.41 |
| `depth` $= 36$ | 2.08 | 12.37 | 75.51 | 1.30 | 1.49 | 12.24 | 0.52 | 0.40 |
| `width` $= 384$ | 0.28 | 16.08 | 76.01 | 1.40 | 1.59 | 11.31 | 0.49 | 0.39 |
| `width` $= 768$ | 0.52 | 14.32 | 77.97 | 1.33 | 1.55 | 12.62 | 0.51 | 0.40 |
| `width` $= 1536^\star$ | 1.44 | 13.01 | 81.23 | 1.29 | 1.50 | 12.42 | 0.52 | 0.41 |
| $k_{\mathrm{convFF}} = 1^\star$ | 1.44 | 13.01 | 81.23 | 1.29 | 1.50 | 12.42 | 0.52 | 0.41 |
| $k_{\mathrm{convFF}} = 3$ | 2.35 | 13.87 | 78.74 | 1.38 | 1.56 | 11.71 | 0.49 | 0.40 |

Table 19: Results on choice of sampler and number of sampling steps using AudioCaps test set. We used the main ETTA model trained for 1M steps and $w_{\mathrm{cfg}} = 3$.

| Sampler | Steps | NFE | $FD_P\downarrow$ | $FD_O\downarrow$ | $KL_S\downarrow$ | $KL_P\downarrow$ | $IS_P\uparrow$ | $CL_L\uparrow$ | $CL_M\uparrow$ |
|---|---|---|---|---|---|---|---|---|---|
| Heun | 100 | 199 | 12.09 | 79.60 | 1.20 | 1.41 | 13.57 | 0.55 | 0.42 |
| Heun | 50 | 99 | **12.00** | **79.24** | 1.19 | 1.41 | 13.61 | 0.55 | 0.43 |
| Heun | 25 | 49 | 12.20 | 80.06 | **1.18** | 1.40 | 13.64 | 0.55 | 0.43 |
| Heun | 10 | 19 | 12.22 | 85.27 | 1.21 | 1.40 | 13.23 | 0.55 | 0.43 |
| Heun | 5 | 9 | 13.27 | 97.45 | 1.27 | 1.43 | 12.22 | 0.52 | 0.42 |
| Euler | 200 | 200 | 12.26 | 79.77 | 1.19 | 1.41 | 13.56 | 0.55 | 0.43 |
| Euler | 100 | 100 | 12.10 | 80.67 | **1.18** | 1.42 | **13.90** | 0.55 | 0.43 |
| Euler | 50 | 50 | **11.83** | 81.49 | **1.18** | **1.39** | 13.45 | 0.55 | 0.43 |
| Euler | 20 | 20 | 12.36 | 90.85 | 1.19 | **1.39** | 13.15 | 0.54 | 0.42 |
| Euler | 10 | 10 | 14.74 | 112.65 | 1.31 | 1.43 | 11.46 | 0.50 | 0.42 |

including the solver, number of function evaluations, and classifier-free guidance.

> Heun sampler is better than Euler at lower NFE under all metrics. Increasing $w_{\mathrm{cfg}}$ improves most objective metrics (KL, IS, and CL) except for FD.

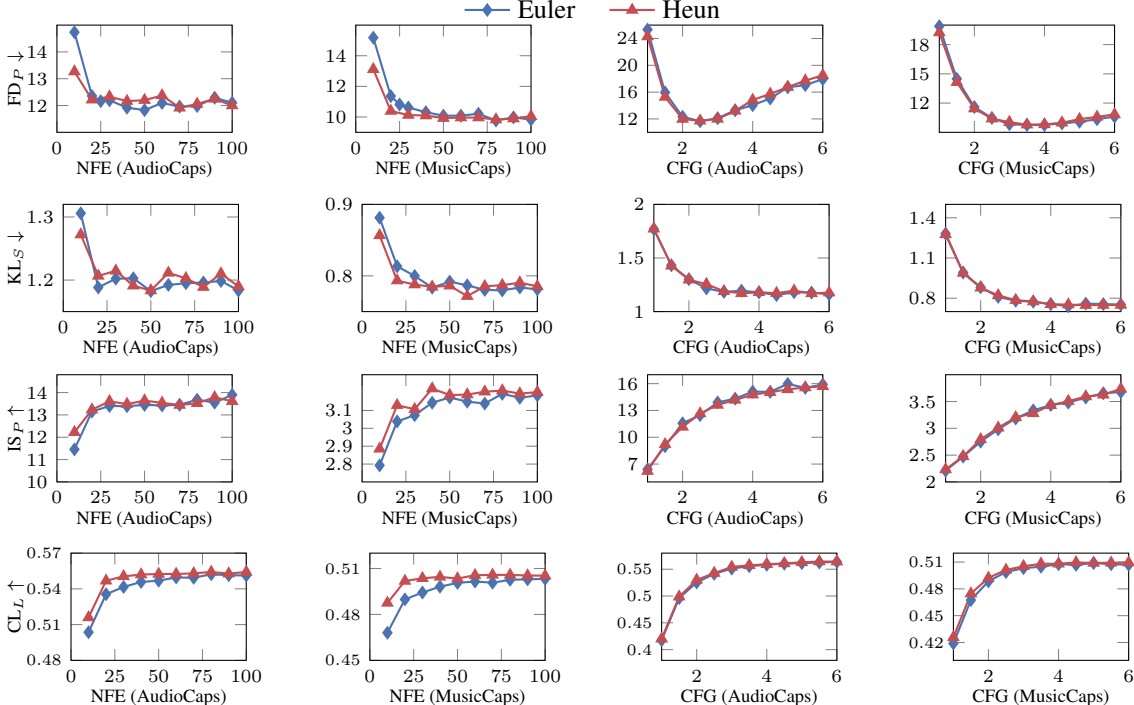

Figure 5: The effect of different sampling methods on the generation quality of ETTA on AudioCaps and MusicCaps. We investigate both Euler and Heun solvers. NFE: number of function evaluations. CFG: classifier-free guidance scale.

Table 20: List of imaginative captions used to generate creative audio.

| Caption |
| --- |
| A hip-hop track using sounds from a construction site—hammering nails as the beat, drilling sounds as scratches, and metal clanks as rhythm accents. |
| A saxophone that sounds like meowing of cat. |
| A techno song where all the electronic sounds are generated from kitchen noises—blender whirs, toaster pops, and the sizzle of cooking. |
| Dogs barking, birds chirping, and electronic dance music. |
| Dog barks a beautiful and fast-paced folk melody while several cats sing chords while meowing. |
| A time-lapse of a city evolving over a thousand years, represented through shifting musical genres blending seamlessly from ancient to futuristic sounds. |
| An underwater city where buildings hum melodies as currents pass through them, accompanied by the distant drumming of bioluminescent sea creatures. |
| A factory machinery that screams in metallic agony. |
| A lullaby sung by robotic voices, accompanied by the gentle hum of electric currents and the soft beeping of machines. |
| A soundscape with a choir of alarm siren from an ambulance car but to produce a lush and calm choir composition with sustained chords. |
| The sound of ocean waves where each crash is infused with a musical chord, and the calls of seagulls are transformed into flute melodies. |
| Mechanical flowers blooming at dawn, each petal unfolding with a soft chime, orchestrated with the gentle ticking of gears. |
| The sound of a meteor shower where each falling star emits a unique musical note, creating a celestial symphony in the night sky. |
| A clock shop where the ticking and chiming of various timepieces synchronize into a complex polyrhythmic composition. |
| An enchanted library where each book opened releases sounds of its story—adventure tales bring drum beats, romances evoke violin strains. |
| A rainstorm where each raindrop hitting different surfaces produces unique musical pitches, forming an unpredictable symphony. |
| A carnival where the laughter of children and carousel music intertwine, and the sound of games and rides blend into a festive overture. |
| A futuristic rainforest where holographic animals emit digital soundscapes, and virtual raindrops produce glitchy electronic rhythms. |
| An echo inside a cave where droplets of water create a cascading xylophone melody, and bats' echolocation forms ambient harmonies. |
| A steampunk cityscape where steam engines puff in rhythm, and metallic gears turning produce mechanical melodies. |

**Creative captions**   Table 20 contains the creative captions for subjective evaluation.

# E. Extended Evaluations

In this section, we present an extended model evaluation including additional out-of-distribution (OOD) benchmark on SongDescriber dataset (Manco et al., 2023). Table 21 and 22 show an extended ablation study on the choice of training dataset. Table 23 includes the OOD evaluation results compared to baseline models. ETTA achieves the lowest $FD_O$ and $FD_P$ as well as the highest CLAP scores compared to baseline models. With only synthetic captions, ETTA matches the REL of Stable Audio Open which is trained on numerous music data and captions. Table 24 compares the AF-AudioSet and AF-Synthetic training sets on the OOD benchmark. Results indicate that ETTA trained with the larger AF-Synthetic leads to significantly better OOD generation results than the smaller AF-AudioSet.

# F. Mixed or Negative Results

In this section, we document additional directions we explored when building ETTA inspired by previous work, but resulted in mixed or worse results in our study. Our goal is not to claim that the methods described below don't work; again, we aim to provide a holistic understanding of design choices commonly found in the TTA literature and speculate that these have been ineffective specific to our experimental setup. We believe that below methods we explored hold the potential to improve results further in future work.

**Pretraining TTA with audio inpainting**   This experiment is inspired by SpeechFlow (Liu et al., 2023a) that presented improvement of various speech tasks (e.g., Text-to-Speech (TTS)) by pretraining the flow matching model with an inpainting task using unlabeled data.

We follow the masking method in (Liu et al., 2023a) and concatenate the masked feature with the noisy input. Note that we do not feed the masked feature to cross-attention input, so the cross-attention parameters are not activated during pretraining. We pretrain the model with this inpainting task for 700k steps. Then, we reset the first input projection layer of DiT and optimizer, and switch to the main TTA task starting from the pretrained weight. We observe that the training loss starts much lower for the pretrained model.

Table 25 summarizes the benchmark results with or without the inpainting as pretraining task. We find that the result is mixed where AudioCaps result worsened and some MusicCaps metrics improved such as FD and IS. We speculate that the pretrained weight focuses more on the music signal because of our unlabeled audio collection has a higher proportion of music compared to speech. We also conjecture that the result would be different if we use the masked feautre to the cross-attention input in pretraining stage instead of concatenation.

> Pretraining with the audio inpainting task produces mixed results, possibly due to data imbalance or sub-optimal implementation details.

While the current experimental setup did not bring positive result, we believe that introducing multiple tasks into a single model will enable a generalist model. We leave exploring alternative ways to ingest the inpainting task into better TTA to future work.

**Choice of Text Encoder**   Many previous works have implemented different text encoders for TTA, but the results are mixed. Researchers have experimented with various models such as BERT, T5, and CLAP to find the optimal text encoder for improving TTA result (Liu et al., 2023b; 2024; Ghosal et al., 2023; Melechovsky et al., 2023; Majumder et al., 2024; Huang et al., 2023c;a). We also explore the text encoder choice in a controlled environment, where we train multiple models with different text encoders. We consider `T5-base`, `T5-large`, `FLAN-T5-base`, and `FLAN-T5-large`. In addition, we experiment with a dual text encoder setup (Huang et al., 2023a; Liu et al., 2024; Haji-Ali et al., 2024) by using CLAP as additional global text embedding. We use a different CLAP checkpoint (LAION's `music_audioset_epoch_15_esc_90.14`) to the benchmark CLAP models ($CL_L$ and $CL_M$) to rule out a possibility of inflated result from the same representation. In this experiment, we start training with a pretrained weight from the inpainting task for 700k training steps, [11] and trained each model with different text encoder for 300k steps.

Table 26 summarizes the result on different text encoder choices evaluated on AudioCaps. Unfortunately, we were not able to discover noticeably better choice compared to others. Nevertheless, we find interesting observations: 1) `FLAN-T5-base` scores relatively better than `T5-base` for $FD_O$, but the opposite can be observed for other metrics such as $KL_S$. 2) for our setup, we have not found strong evidence that dual text encoder with CLAP is better; it worsened most metrics except for `FLAN-T5-large`. 3) larger T5 encoder may not necessarily be better in improving results, where `base` model generally scored better metrics than `large` model. `T5-large` showed surprisingly worse result compared to others for two independent training runs (with or without CLAP). While this seems counter-intuitive, it also suggests that the optimal choice of text encoder would depend on other factors such as training dataset and the main TTA model capacity at hand.

---

[11] We launched this experiment based on the preliminary observation of the lower training loss. We speculate that the observation would not change if we train the models from scratch.

Table 21: Ablation study on the results of ETTA trained on different datasets (evaluated on AudioCaps).

| Dataset (million captions) | $FD_P\downarrow$ | $FD_O\downarrow$ | $KL_S\downarrow$ | $KL_P\downarrow$ | $IS_P\uparrow$ | $CL_L\uparrow$ | $CL_M\uparrow$ |
|---|---|---|---|---|---|---|---|
| AudioCaps (0.05) | **22.60** | 95.99 | **1.49** | **1.63** | **6.73** | **0.48** | **0.35** |
| TangoPromptBank (1.21) | 33.44 | **77.07** | 2.39 | 2.72 | 4.64 | 0.29 | 0.27 |
| AF-AudioSet (0.16) | 25.06 | 108.31 | 1.81 | 2.01 | 6.32 | 0.42 | 0.34 |
| AF-Synthetic (1.35) | 28.46 | 89.60 | 1.99 | 2.21 | 5.64 | 0.37 | 0.32 |

Table 22: Ablation study on the results of ETTA trained on different datasets (evaluated on MusicCaps).

| Dataset (million captions) | $FD_P\downarrow$ | $FD_O\downarrow$ | $KL_S\downarrow$ | $KL_P\downarrow$ | $IS_P\uparrow$ | $CL_L\uparrow$ | $CL_M\uparrow$ |
|---|---|---|---|---|---|---|---|
| AudioCaps (0.05) | 76.14 | 279.44 | 3.20 | 3.63 | 2.05 | 0.12 | 0.27 |
| TangoPromptBank (1.21) | 24.72 | **86.17** | 1.73 | 2.02 | 2.27 | 0.35 | 0.38 |
| AF-AudioSet (0.16) | 21.40 | 107.00 | 1.45 | 1.52 | **2.36** | 0.40 | **0.44** |
| AF-Synthetic (1.35) | **21.59** | 92.30 | **1.41** | **1.51** | 2.20 | **0.41** | **0.44** |

Table 23: Additional results of ETTA compared to SOTA baselines on SongDescriber.

| Model | $FD_P\downarrow$ | $FD_O\downarrow$ | $KL_S\downarrow$ | $KL_P\downarrow$ | $IS_P\uparrow$ | $CL_L\uparrow$ | $CL_M\uparrow$ | $OVL\uparrow$ | $REL\uparrow$ |
|---|---|---|---|---|---|---|---|---|---|
| Ground Truth | – | – | – | – | 1.88 | 0.48 | 0.38 | $4.37\pm0.07$ | $4.26\pm0.09$ |
| AudioLDM2 | 16.02 | 335.37 | 0.74 | 0.78 | 1.93 | 0.42 | 0.45 | – | – |
| AudioLDM2-large | 10.50 | 324.38 | **0.67** | **0.75** | 1.95 | **0.44** | 0.48 | $3.37\pm0.10$ | $3.38\pm0.11$ |
| TANGO-AF | 21.49 | 233.32 | 0.79 | 0.88 | 1.96 | 0.43 | 0.44 | $3.32\pm0.10$ | $3.36\pm0.11$ |
| Stable Audio Open | 34.76 | 129.88 | 0.99 | 1.01 | **2.19** | 0.42 | 0.47 | $\mathbf{3.92}\pm0.10$ | $\mathbf{3.80}\pm0.10$ |
| *ETTA* | **9.98** | **95.66** | 0.80 | 0.76 | 2.15 | **0.44** | **0.53** | $\underline{3.70}\pm0.09$ | $\underline{3.79}\pm0.10$ |

Table 24: Ablation study on the results of ETTA trained on different datasets (evaluated on SongDescriber).

| Dataset (million captions) | $FD_P\downarrow$ | $FD_O\downarrow$ | $KL_S\downarrow$ | $KL_P\downarrow$ | $IS_P\uparrow$ | $CL_L\uparrow$ | $CL_M\uparrow$ |
|---|---|---|---|---|---|---|---|
| AF-AudioSet (0.16) | 12.97 | 125.16 | 1.03 | 0.89 | **2.36** | 0.41 | 0.50 |
| AF-Synthetic (1.35) | **10.29** | **104.16** | **0.80** | **0.76** | 2.06 | **0.43** | **0.51** |

Table 25: Results on pretraining with the audio inpainting task vs. training from scratch. In either case, ETTA is trained on the TTA task for 250k steps.

| Dataset | Pretrain | $FD_P\downarrow$ | $FD_O\downarrow$ | $KL_S\downarrow$ | $KL_P\downarrow$ | $IS_P\uparrow$ | $CL_L\uparrow$ | $CL_M\uparrow$ |
|---|---|---|---|---|---|---|---|---|
| AudioCaps | ✓ | 12.90 | 90.77 | 1.40 | 1.54 | 11.87 | 0.49 | 0.40 |
| AudioCaps | ✗ | 13.01 | 81.23 | 1.29 | 1.50 | 12.42 | 0.52 | 0.41 |
| MusicCaps | ✓ | 10.87 | 81.19 | 0.91 | 1.10 | 3.03 | 0.51 | 0.50 |
| MusicCaps | ✗ | 12.15 | 96.46 | 0.88 | 1.08 | 2.93 | 0.51 | 0.52 |

Table 26: Effects of different text encoders in ETTA (evaluated on AudioCaps). We initialize weights from a checkpoint that is pretrained on the audio inpainting task for 700k steps and train each model on the TTA task for 300k steps.

| $Enc_{T5}$ | $Enc_{clap}$ | $FD_P\downarrow$ | $FD_O\downarrow$ | $KL_S\downarrow$ | $KL_P\downarrow$ | $IS_P\uparrow$ | $CL_L\uparrow$ | $CL_M\uparrow$ |
|---|---|---|---|---|---|---|---|---|
| T5-base | ✗ | 12.93 | 91.81 | 1.42 | 1.57 | 12.57 | 0.49 | 0.40 |
| T5-base | ✓ | 12.87 | 85.40 | 1.42 | 1.58 | 11.91 | 0.48 | 0.39 |
| T5-large | ✗ | 16.80 | 213.71 | 1.61 | 1.63 | 11.91 | 0.45 | 0.36 |
| T5-large | ✓ | 18.73 | 219.93 | 1.67 | 1.74 | 9.44 | 0.43 | 0.35 |
| FLAN-T5-base | ✗ | 13.01 | 80.67 | 1.50 | 1.61 | 13.26 | 0.48 | 0.40 |
| FLAN-T5-base | ✓ | 13.51 | 84.08 | 1.50 | 1.62 | 11.37 | 0.47 | 0.39 |
| FLAN-T5-large | ✗ | 15.94 | 103.15 | 1.63 | 1.70 | 10.60 | 0.45 | 0.38 |
| FLAN-T5-large | ✓ | 13.28 | 81.64 | 1.52 | 1.63 | 11.80 | 0.47 | 0.39 |

> No single text encoder consistently outperformed others. The effectiveness of text encoders seems to depend on specific metrics and setup. Larger text encoders do not always lead to better results.

**Autoguidance** Recently, (Karras et al., 2024) showed that the improvement in perceptual quality of CFG stems from its ability to eliminate unlikely outlier samples, but it may reduce diversity from over-emphasis. They proposed a new way of guiding the model, called *autoguidance*, that uses a bad version of the same model (either by under-

Table 27: Results on AutoGuidance (evaluated on AudioCaps). We use our best 1.44B ETTA model (trained for 1M steps). $Model_{ag}$ denotes the bad model used for AutoGuidance. Same: same 1.44B model architecture as ETTA. XS: the smallest 0.28B model using `width=384`.

| $Model_{ag}$ (steps) | $w_{cfg}$ | $w_{ag}$ | $FD_P\downarrow$ | $FD_O\downarrow$ | $KL_S\downarrow$ | $KL_P\downarrow$ | $IS_P\uparrow$ | $CL_L\uparrow$ | $CL_M\uparrow$ |
|---|---|---|---|---|---|---|---|---|---|
| – | 1 | – | 25.33 | 93.03 | 1.77 | 2.00 | 6.41 | 0.42 | 0.34 |
| XS (50$k$) | 1 | 2 | 14.51 | 86.14 | 1.63 | 1.73 | 9.10 | 0.51 | 0.38 |
| XS (50$k$) | 3 | 2 | 12.15 | 80.24 | 1.20 | **1.41** | 13.64 | **0.55** | **0.43** |
| XS (100$k$) | 1 | 2 | 14.19 | 83.52 | 1.63 | 1.72 | 8.54 | 0.50 | 0.38 |
| XS (100$k$) | 3 | 2 | 14.15 | 94.08 | 1.37 | 1.49 | 13.83 | 0.55 | 0.42 |
| Same (100$k$) | 1 | 2 | 16.49 | 91.79 | 1.58 | 1.78 | 7.63 | 0.48 | 0.37 |
| Same (100$k$) | 3 | 2 | 12.72 | 81.85 | 1.27 | 1.50 | 13.80 | 0.56 | 0.42 |
| – | 3 | – | **12.10** | **80.67** | **1.18** | 1.42 | **13.90** | 0.55 | 0.43 |

Table 28: Results on AutoGuidance (evaluated on MusicCaps). We use our best 1.44B ETTA model (trained for 1M steps). We report the results using the best combination according to Table 27.

| $Model_{ag}$ | $w_{cfg}$ | $w_{ag}$ | $FD_P\downarrow$ | $FD_O\downarrow$ | $KL_S\downarrow$ | $KL_P\downarrow$ | $IS_P\uparrow$ | $CL_L\uparrow$ | $CL_M\uparrow$ |
|---|---|---|---|---|---|---|---|---|---|
| – | 1 | – | 19.89 | 101.15 | 1.28 | 1.43 | 2.21 | 0.42 | 0.46 |
| XS (50$k$) | 1 | 2 | 13.17 | 104.43 | 1.12 | 1.32 | 2.59 | 0.48 | 0.49 |
| XS (50$k$) | 3 | 2 | 9.83 | **97.63** | **0.78** | **1.03** | 3.19 | **0.50** | **0.53** |
| – | 3 | – | **9.82** | 98.19 | **0.78** | **1.03** | **3.18** | **0.50** | **0.53** |

training and/or with smaller model) that increases diversity while ensuring high-quality output as follows (omitting the condition $c$ for brevity):

$$v_\theta(x_t, t) = v_{\theta_{ag}}(x_t, t) + w_{ag} \cdot (v_\theta(x_t, t) - v_{\theta_{ag}}(x_t, t)),$$

where $\theta_{ag}$ denotes a bad model and $w_{ag}$ is the scale for autoguidance. Same as CFG, $w_{ag} = 1$ disables the guidance and $w_{ag} > 1$ amplifies the main model's prediction.

We conducted experiments applying autoguidance to evaluate its effectiveness to our TTA setup. The results are in Tables 27 and 28. From our grid search of $w_{ag}$ from 1 to 2.5 with 0.25 interval, $w_{ag} = 2$ provided the best possible metrics.

Subjectively, we observed that while autoguidance could produce more diverse audio samples corroborating (Karras et al., 2024), but these samples sometimes lacked realism. We find that the method is sensitive to the choice of the bad model and its guidance scale $w_{ag}$. In terms of improving benchmark results, despite our best efforts and various combinations including different bad models (either under-trained versions or smaller models) and guidance scales, we were unable to identify a setup that clearly outperforms plain CFG with $w_{cfg} = 3$. Similar benchmark metrics could only be achieved by combining both CFG and autoguidance, but at an increased cost with 2x NFE.

We conjecture that our search space may have been incomplete. However, we do observe noticeable increase in diversity from autoguidnace where the same ETTA checkpoint can sometimes generate even more "interesting" samples,

so we believe autoguidance holds its potential towards creativity. We leave exploring recently proposed methods for sampling from the model for better TTA results in future work.

> AutoGuidance increases diversity but does not consistently outperform CFG in objective metrics. It shows potential for diversity, though its effectiveness is sensitive to model and scale choices.

**Min-SNR-$\gamma$ training strategy** (Hang et al., 2023) We use $\gamma = 5$ per convention, and trained ETTA-DiT for 250k steps with the $v$-diffusion objective. Results can be found in Table 29 and 30 ($w_{cfg} = 3$). Most metrics became worse, but and $FD_O$ and $IS_P$ are slightly better in music generation.

**CFG on a limited interval** (Kynkäänniemi et al., 2024) We remove CFG for the initial 40% of diffusion steps and then apply CFG for the remaining 60% steps. We inference and evaluate both with and without AutoGuidance. Results can be found in Table 31 and 32. Most metrics became worse, but is slightly better in audio generation using the small (XS) model.

Table 29: Additional Results on training strategies (evaluated on AudioCaps).

| Ablation | FD$_P$↓ | FD$_O$↓ | KL$_S$↓ | KL$_P$↓ | IS$_P$↑ | CL$_L$↑ | CL$_M$↑ |
|---|---|---|---|---|---|---|---|
| Stable Audio Open | 38.27 | 105.88 | 2.23 | 2.32 | 12.09 | 0.35 | 0.34 |
| + AF-Synthetic | 18.50 | 86.13 | 1.58 | 1.74 | 14.96 | 0.47 | 0.40 |
| + ETTA-DiT | 16.43 | 90.26 | 1.29 | 1.47 | 14.49 | 0.53 | 0.42 |
| + Min-SNR-$\gamma$ ($\gamma = 5$) | 18.00 | 100.86 | 1.36 | 1.56 | 13.85 | 0.52 | 0.40 |

Table 30: Additional Results on training strategies (evaluated on MusicCaps).

| Ablation | FD$_P$↓ | FD$_O$↓ | KL$_S$↓ | KL$_P$↓ | IS$_P$↑ | CL$_L$↑ | CL$_M$↑ |
|---|---|---|---|---|---|---|---|
| Stable Audio Open | 36.42 | 127.20 | 1.32 | 1.56 | 2.93 | 0.48 | 0.49 |
| + AF-Synthetic | 14.59 | 103.59 | 1.00 | 1.20 | 3.19 | 0.50 | 0.52 |
| + ETTA-DiT | 12.48 | 98.19 | 0.82 | 1.06 | 3.30 | 0.50 | 0.52 |
| + Min-SNR-$\gamma$ ($\gamma = 5$) | 13.04 | 97.44 | 0.89 | 1.12 | 3.66 | 0.50 | 0.50 |

Table 31: Additional Results on Guidance on Limited Interval (evaluated on AudioCaps).

| Model$_{ag}$ (steps) | $w_{cfg}$ | $w_{ag}$ | FD$_P$↓ | FD$_O$↓ | KL$_S$↓ | KL$_P$↓ | IS$_P$↑ | CL$_L$↑ | CL$_M$↑ |
|---|---|---|---|---|---|---|---|---|---|
| – | 1 | – | 25.33 | 93.03 | 1.77 | 2.00 | 6.41 | 0.42 | 0.34 |
| XS ($50k$) | 3 | 2 | 12.15 | 80.24 | 1.20 | **1.41** | 13.64 | **0.55** | **0.43** |
| + CFG @ [0, 0.6] | 3 | 2 | **11.74** | 89.18 | 1.44 | 1.59 | 10.59 | 0.54 | 0.40 |
| – | 3 | – | 12.10 | **80.67** | **1.18** | 1.42 | **13.90** | **0.55** | **0.43** |
| + CFG @ [0, 0.6] | 3 | – | 16.13 | 93.28 | 1.45 | 1.66 | 8.35 | 0.48 | 0.38 |

Table 32: Additional Results on Guidance on Limited Interval (evaluated on MusicCaps).

| Model$_{ag}$ | $w_{cfg}$ | $w_{ag}$ | FD$_P$↓ | FD$_O$↓ | KL$_S$↓ | KL$_P$↓ | IS$_P$↑ | CL$_L$↑ | CL$_M$↑ |
|---|---|---|---|---|---|---|---|---|---|
| – | 1 | – | 19.89 | 101.15 | 1.28 | 1.43 | 2.21 | 0.42 | 0.46 |
| XS ($50k$) | 3 | 2 | 9.83 | **97.63** | **0.78** | **1.03** | 3.19 | **0.50** | **0.53** |
| + CFG @ [0, 0.6] | 3 | 2 | 11.40 | 102.66 | 1.02 | 1.26 | 2.79 | 0.49 | 0.50 |
| – | 3 | – | **9.82** | 98.19 | **0.78** | **1.03** | **3.18** | **0.50** | **0.53** |
| + CFG @ [0, 0.6] | 3 | – | 16.43 | 100.18 | 1.12 | 1.27 | 2.37 | 0.44 | 0.48 |

# G. Vocoder/Autoencoder Reconstruction Results

Table 33 and 34 show objective results of our VAE we used (ETTA-VAE) in this work. Our 44kHz stereo VAE is identical to the one used in Stable Audio Open (Evans et al., 2024c), but trained from scratch using our large-scale unlabeled audio collection based on public datasets. We also attach BigVGAN-v2 (Lee et al., 2023), the state-of-the-art mel spectrogram vocoder in 44kHz mono, as a reference of waveform reconstruction quality from the models.

Despite being 4x lower in latent frame rate (21.5Hz) compared to conventional mel spectrogram vocoder (86Hz), ETTA-VAE shows competitive reconstruction quality. It matches the quality of Stable Audio Open-VAE on music data (MUSDB18-HQ (Rafii et al., 2017)) and outperforms on speech data (LibriTTS (Zen et al., 2019)), because our dataset contains considerably high portion of speech signals.

> Our ETTA-VAE matches or exceeds the reconstruction quality of Stable Audio Open's VAE. This is because we use larger-scale public audio datasets.

Table 33: Comparison of waveform vocoder/auto-encoder on LibriTTS (dev-clean and dev-other).

| Model | Framerate | PESQ↑ | UTMOS↑ | ViSQOL↑ | M-STFT↓ | SI-SDR↑ |
|---|---|---|---|---|---|---|
| Ground Truth | - | 4.64 | 3.86 | 4.73 | – | – |
| BigVGAN-v2 | 86 Hz | 4.14 | 3.73 | 4.69 | 0.71 | -7.86 |
| Stable Audio Open-VAE | 21.5 Hz | 2.75 | 3.13 | 4.31 | 1.00 | 7.15 |
| ETTA-VAE | 21.5 Hz | 3.18 | 3.76 | 4.37 | 0.79 | 9.92 |

Table 34: Comparison of waveform vocoder/autoencoder on MUSDB18-HQ test set.

| Model | Framerate | ViSQOL↑ | M-STFT↓ | SI-SDR↑ |
|---|---|---|---|---|
| Ground Truth | - | 4.73 | – | – |
| BigVGAN-v2 | 86 Hz | 4.63 | 0.94 | -22.06 |
| Stable Audio Open-VAE | 21.5 Hz | 4.25 | 1.00 | 9.34 |
| ETTA-VAE | 21.5 Hz | 4.27 | 0.95 | 10.59 |

