# OpenReview forum: "ETTA: Elucidating the Design Space of Text-to-Audio Models"
_ICML.cc/2025/Conference — ICML 2025 poster_

### Official Review · Reviewer_gQwi · 2025-03-12

**Overall Recommendation:** 4

**Summary:**

This paper presents a high-quality audio-caption dataset containing 1.35M pairs of data, named as AF-Synthetic. The dataset is established through the state-of-the-art (SoTA) audio-language model, Audio Flamingo. In addition, the paper introduces a text-to-audio system based on the Diffusion Transformer (DiT) based Latent Diffusion Model. The experimental results demonstrate that the proposed system, named Elucidated Text-To-Audio (ETTA), achieves SoTA performance on both the natural audio and music.

**Claims And Evidence:**

The authors claim the effectiveness of the proposed dataset by demonstrating the SoTA performance of the ETTA system trained on AF-Synthetic. However, the paper lacks sufficient experiments directly comparing the dataset's contribution, such as showing how other SoTA models, such like Tango and AudioLDM, perform when trained with AF-Synthetic.

**Essential References Not Discussed:**

None

**Experimental Designs Or Analyses:**

The paper lacks more experiments to illustrate the effectiveness of AF-Synthetic dataset on other systems.
In addition, as an audio-language dataset, experiments that present the contribution of the dataset on other tasks, such as audio retrieval and audio captioning can enhance the significant usefulness of the proposed dataset.

**Methods And Evaluation Criteria:**

The main weakness of the paper is that using audio-language model, such as Audio Flamingo to generate the caption making the result more like a conversation, not just normal captions. Most of the captions will begin with ”the audio consists”, “the audio features”,”There is a”. Which is actually different to the real-world cases.

**Other Comments Or Suggestions:**

Please see questions.

**Other Strengths And Weaknesses:**

The proposed system, named as ETTA, more like a model applying Stable Audio as the backbone. Not too much novel techniques are proposed in this system.

**Questions For Authors:**

Overall, this is a high-quality paper and the idea of creating an audio-caption dataset is interesting. I am happy to change the score if the author can solve the following questions or experiments.

Question1: What is the performance of other text-to-audio system that train on AF-Synthetic? Such as AudioLDM, Tango, Make an Audio.

Question2:What is the difference between ETTA and Stable-Audio(Despite the DIT module)?

Question3:Why choose the current CLAP threshold, will the dataset be improved if raises the score?

Question4:Did the audio-language format of the caption lead to some problems? Is there any way to avoid such structure and re-caption into a more formal structure?

Question5:Why the ETTA does not outperform previous models in some MOS metrics?

Question6: Could you also provide some demos on music?

**Relation To Broader Scientific Literature:**

The paper mainly provides a larger-scale audio dataset, which is useful for audio-related tasks

**Theoretical Claims:**

None

---

> ### Author Rebuttal · Authors · 2025-03-31
>
> Thank you for your supportive review. We address your concerns as follows:
>
> **Q: The paper lacks sufficient experiments directly comparing the dataset's contribution, such as showing how other SoTA models, such like Tango and AudioLDM, perform when trained with AF-Synthetic. What is the performance of other text-to-audio systems that train on AF-Synthetic? Such as AudioLDM, Tango, Make an Audio.**
>
> A: We acknowledge your concern; current main results (Table 2 and 3) are structured to represent prior works in their original form by using the official results and/or their public checkpoints.
>
> Stable Audio Open is our most modern and direct baseline, and we did perform the ablation study in AF-Synthetic dataset’s direct impact while keeping the baseline model identical in Table 4 (+AF-Synthetic). Compared to the public Stable Audio Open checkpoint, AF-Synthetic provides significant improvements from the training dataset alone.
>
> In our preliminary study, we also did our best effort to train previous models (other than Stable Audio Open) using AF-Synthetic, but except for Stable Audio Open we have faced difficulties using their original training recipe (e.g. model not converging, loss instabilities, etc.). We speculate these models would need non-trivial optimization in the training recipe to accommodate significantly larger-scale data such as AF-Synthetic.
>
>
> **Q: The main weakness of the paper is that using audio-language model, such as Audio Flamingo to generate the caption making the result more like a conversation, not just normal captions. Most of the captions will begin with ”the audio consists”, “the audio features”,”There is a”. Which is actually different to the real-world cases. + Did the audio-language format of the caption lead to some problems? Is there any way to avoid such structure and re-caption into a more formal structure?**
>
> A: Thank you for asking this important question. While AF-Synthetic captions mostly start with a conversational prefix from the audio language model (Audio Flamingo), we empirically find that the model is robust in generating good quality samples using plain captions, as evidenced by our main benchmarks that use the original captions (Tables 2, 3, and 22) along with the OOD imaginative prompts in the demo webpage.
>
> We also tried using an external LLM (Nemotron-340B [1]) to rephrase and shorten the AF-Synthetic captions to be more caption style (like AudioCaps). We found ETTA has similar quality when trained on this rephrased dataset.
>
> [1] Adler, Bo, et al. "Nemotron-4 340b technical report." arXiv preprint arXiv:2406.11704 (2024).
>
> **Q: What is the difference between ETTA and Stable-Audio(Despite the DIT module)?**
>
> A: Other than the improved DiT backbone, ETTA uses an improved VAE, a better training objective (OT-CFM w/ logit-normal t-sampling), and is trained on AF-Synthetic.
>
> **Q: Why choose the current CLAP threshold, will the dataset be improved if raises the score?**
>
> A: The CLAP threshold >= 0.45 is set following a suggestion in existing work [2]. Improving the CLAP threshold will retain higher quality subset at a cost of reduction of the dataset size.
>
> [2] Kong, Zhifeng, et al. "Improving text-to-audio models with synthetic captions." arXiv preprint arXiv:2406.15487 (2024).
>
>
> **Q: Why the ETTA does not outperform previous models in some MOS metrics?**
>
> A: For music, Stable Audio Open is trained on high quality music from Freesound and Free Music Archive (FMA) datasets, but many of these samples are not available as of today. In contrast, ETTA is not trained on any studio-quality music datasets (not even MusicCaps). As a result, ETTA has a slightly worse OVL (which only measures audio quality and does not consider text) on MusicCaps and SongDescriber.
>
> While only trained on synthetic captions, ETTA outperforms Stable Audio Open on the REL (audio-text relevance) score on AudioCaps and MusicCaps (Table 2-3), and matches it on an OOD testset SongDescriber (Table 22).
>
>
>
> **Q: Could you also provide some demos on music?**
>
> A: We have added several MusicCaps benchmark samples to the updated demo webpage. Link: https://anonymous.4open.science/r/etta_demo-778A/index.md

---

> > ### Comment · Reviewer_gQwi · 2025-04-02
> >
> > The author has explained and answered most of my concerns. I have raised my score and good luck with the paper.

---

### Official Review · Reviewer_wC5z · 2025-03-12

**Overall Recommendation:** 4

**Summary:**

This paper proposes ETTA, a state-of-the-art text-to-audio model trained on public data. Its innovations include:
- A new dataset called AF-Synthetic that follows the audio captioning pipeline from AF-AudioSet but scales up to million-scale. This is done by captioning or re-captioning audio in AudioCaps, AudioSet, VGGSound, WavCaps, and LAION-630K datasets.
- A latent diffusion/flow matching model that builds on Stable Audio and makes several architectural changes.
- An analysis on the scalability with respect to model size and training data size.
- An ablation study on diffusion sampler and steps (NFEs).

**Claims And Evidence:**

Yes in general, but I have minor questions about some details.

- The authors claim that "We find the 1.44B model with depth=24 and width=1536 leads to an optimal balance between model size and quality." However, according to Table 6, the 2.08B 36-layer model seems to produce better results. So why is the balance achieved by the 1.44B model optimal?
- The authors use experiments to show that an excessively high CFG scale is suboptimal in terms of FD, claiming the compromised diversity as the reason for such a suboptimality. I found this under-motivated, as CFG could suffer FD penalties via other mechanisms, such as distortion caused by CFG's extrapolation.

**Essential References Not Discussed:**

No.

**Experimental Designs Or Analyses:**

Yes. They are sound.

**Methods And Evaluation Criteria:**

Yes in general. I would like to point out several minor things.

- The authors use CLAP score to show the superiority of AF-Synthetic captions. While I generally trust CLAP and am convinced about the quality of the proposed data captions, I believe it would benefit from a small-scale human listening test to verify AF-Synthetic's superiority. That is, for a small set of audio pieces shared between AF-Synthetic and AudioCaps or WavCaps or Laion-630K, I suggest letting human raters compare the captions in AF-Synthetics and those in the source dataset.
- It would also be nice to include AudioCaps, WavCaps, and Laion-630K in Figure 1 (can be random subsets of these datasets).
- When evaluating the distributional similarity between AF-Synthetic captions and other caption datasets, the authors considered a CLAP-score-based max-similarity metric. An alternative and potentially more straightforward metric would be Frechet embedding distance. That is, we can use a similar approach to computing FAD using a CLAP backbone, except now we use CLAP text embeddings instead of audio embeddings. A smaller distance between two caption sets means more similar captions.
- The authors make several architectural changes to the DiT model (AdaLN, GELU, etc.) How were these modifications decided? Were they added all at once, or one by one? If they were added to the baseline DiT one by one, which modification made the largest impact?
- The authors compare flow-matching models with $v$-prediction diffusion models. To my knowledge, score-prediction diffusion models are also very popular and are known to produce high performance. Has it been considered to compare with them?

**Other Comments Or Suggestions:**

- There are a lot of interesting results buried in the appendix. It would be nice to briefly mention some of them in the main paper body.
- I found it interesting that FLAN-T5-Large is not universally better than T5-Base. Not asking for additional experiments, but do you think this is a "quirk" of this particular model, or is it something likely shared across different types of conditional diffusion/flow matching models?
- In Table 2, there are several models comparable with (and even better than) "ETTA (AudioCaps only)" in some metrics. For example, Make-An-Audio 2, TANGO-AF&AC-FT-AC, TANGO2, and GenAU-L all seem strong. Are these models comparable with "ETTA (AudioCaps only)"? Were they trained on additional data? Would be nice to add an "extra training data" column to that table.

**Other Strengths And Weaknesses:**

**Overall, this paper is comprehensive, thorough, well-written, and worth accepting.** It shares important insights into various dimensions of TTA model design. I had a good laugh when I listened to the demo.

**As mentioned above, several things that can be improved are:**
- Stronger motivations and explanations for architectural changes.
- Human evaluation on the captions of AF-synthetic.
- (Stretch goal) compare with a score-prediction diffusion setup. This paper is already quite comprehensive, and I am aware that training a new model from scratch could be time-consuming and infeasible to complete within the rebuttal time frame. So this is optional.

**If the above can be addressed, I recommend accepting this paper.**

**Questions For Authors:**

- Is there any chance that the AF-Synthetic dataset can be open-sourced?
- What does "Finally, we also sub-sample long audio segments except for music and speech" mean in Section 3.1?
- What does "We switch from prepending to AdaLN timestep embedding and apply AdaLN" mean in Section 3.2? I'm also confused with the footnote "in our preliminary study using stable-audio-tools with its vanilla implementation, switching from prepending to AdaLN resulted in worse results." So AdaLN didn't work with vanilla implementation but worked for ETTA? Why?

**Relation To Broader Scientific Literature:**

See summary.

**Theoretical Claims:**

N/A.

---

> ### Author Rebuttal · Authors · 2025-03-31
>
> Thank you for your supportive review. We address your questions as follows:
>
> **Q: according to Table 6, the 2.08B 36-layer model seems to produce better results. So why is the balance achieved by the 1.44B model optimal?**
>
> A: We observed improvements for the deeper model (2.08B), but the gap narrows with CFG (Table 16 vs. 17). The subjective quality has been similar from the preliminary study. For this work we consider 1.44B a good balance between quality and speed, but we also expect that larger models will become better as we scale up the synthetic dataset.
>
> **Q: CFG could suffer FD penalties via other mechanisms, such as distortion caused by CFG's extrapolation.**
>
> A: This is a very good point. We also notice oversmoothing and/or distortion when CFG is set too high. This can be another source of its impact on FD. We will update the description accordingly to reflect this.
>
> **Q:  I suggest letting human raters compare the captions in AF-Synthetics and those in the source dataset + An alternative and potentially more straightforward metric would be Frechet embedding distance.**
>
> A: Thank you for your suggestion to evaluate the subjective quality of the synthetic caption from AF-Synthetic. We also agree CLAP does provide a good proxy that quantifies the caption quality, as evidenced by ETTA’s generation results by using AF-Synthetic as the training dataset.
>
> We are happy to conduct human evaluation of AF-Synthetic captions. Since it is a very large scale, we will randomly sample a subset of captions and report human results in the final version.
>
> The purpose of the max-sim metric is to assess to which extent we generate novel captions in AF-Synthetic. A max-sim close to 1.0 indicates that generated captions are copied from the source captions, which is not desired. We show in Figure 2 that max-sim is not close to 1.0, suggesting the generated captions are novel and diverse (see Table 9 and 10 for some qualitative examples).
>
>
> **Q: It would also be nice to include AudioCaps, WavCaps, and Laion-630K.**
>
> A: Nice suggestion. We will include the full results in the paper.
>
> **Q: The authors make several architectural changes to the DiT model (AdaLN, GELU, etc.) How were these modifications decided? Were they added all at once, or one by one? + AdaLN didn't work with vanilla implementation but worked for ETTA? Why?**
>
> A: ETTA-DiT modifications have been made entirely, inspired by recent best practices. We conjecture adding AdaLN to both self-attention & cross-attention input with unbounded gating provides stronger conditioning signals.
>
> We provide training loss curves between Stable Audio-DiT vs. ETTA-DiT (Figure-Rebuttal-1), link: https://anonymous.4open.science/r/etta_demo-778A/index.md . ETTA-DiT provides better convergence whereas Stable Audio Open-DiT plateaus early.
>
> **Q: To my knowledge, score-prediction diffusion models are also very popular and are known to produce high performance. Has it been considered to compare with them?**
>
> We did not consider the score (epsilon) prediction diffusion but used velocity (v) prediction in this work, as v-prediction has been the default choice of our baseline (Stable Audio Open). To the best of the author's knowledge, v-prediction has been favored by practitioners due to its better stability during training. We also find that OT-CFM provides even better stability.
>
>
> **Q:  Make-An-Audio 2, TANGO-AF&AC-FT-AC, TANGO2, and GenAU-L all seem strong. Are these models comparable with "ETTA (AudioCaps only)"? Were they trained on additional data?**
>
> A: Baselines models are from their own official results and/or the public checkpoints trained with different datasets. ETTA (AudioCaps only) in Table 2 is also to illustrate improved results by fine-tuning ETTA on AudioCaps resuming from a pre-trained ETTA with AF-Synthetic. We’ll add the extra training data column to the main tables.
>
> **Q: Is there any chance that the AF-Synthetic dataset can be open-sourced?**
>
> A: We will release the model code and data preparation methods to reproduce the result of ETTA models.
>
> **Q: I found it interesting that FLAN-T5-Large is not universally better than T5-Base.**
>
> A: Thank you for mentioning this. While we are not drawing a conclusive claim, it suggests that an optimal choice may also depend on other factors (i.e. “quirks”). We also speculate that another possible reason is the difference in text embedding variance between the encoders [1].
>
> [1] Xie, Enze, et al. "Sana: Efficient high-resolution image synthesis with linear diffusion transformers." arXiv preprint arXiv:2410.10629 (2024).
>
> **Q: What does "Finally, we also sub-sample long audio segments except for music and speech" mean in Section 3.1?**
>
> A: For long sound (non-music or speech) data, we found most of them to be homogenized (e.g. an hour of continuous engine sound). For these samples, we sub-sample a few ten-second segments instead of adding the entire 360 segments to AF-Synthetic.

---

> > ### Comment · Reviewer_wC5z · 2025-04-05
> >
> > Thanks to the authors for the response. My questions have been resolved, and I believe the paper is worth accepting.

---

### Official Review · Reviewer_okEU · 2025-03-13

**Overall Recommendation:** 3

**Summary:**

This paper explores the design space affecting text-to-audio generation models. Specifically, the authors analyze the effects of dataset quality and scale, architectural and training/inference design choices, and sampling methods during inference. For this purpose, a new large-scale synthetic dataset, AF-Synthetic, is constructed, and several architectural ablations are performed based on existing text-to-audio models. Extensive experiments indicate that both dataset size and quality are important, with quality having more impact. Additionally, improving DiT and applying OT-CFM result in more stable training and improved outcomes.

**Claims And Evidence:**

- The authors claim that the proposed AF-Synthetic significantly improves text-to-audio generation quality. While Table 4 shows improvement for the existing Stable Audio Open trained on AF-Synthetic, Table 5 indicates that performance with AF-AudioSet is comparable or even superior. Similarly, Table 20 suggests using AF-AudioSet is generally preferable. Although the reviewer acknowledges the authors' efforts in building a large dataset, it remains challenging to conclude that AF-Synthetic is crucial for model improvement.

- The authors also argue that their proposed design choices effectively enhance model performance. Tables 5, 12, and 13 demonstrate improvements with AF-Synthetic. However, simply adding components does not fully support the effectiveness of each design choice, as results fluctuate. Additional analyses—such as human perception studies, loss curves, or convergence times—would better illustrate each component's impact.-

**Essential References Not Discussed:**

N/A

**Experimental Designs Or Analyses:**

- Table 4 makes it difficult to interpret the contributions of proposed components, indicating that dataset scale has the largest effect, while other components yield mixed results. Clarifying effects via human evaluations, additional datasets, or deeper analyses would help. Tables 12 and 13 similarly show mixed results, suggesting dataset choice as the most influential factor.

- Why are the results of Stable Audio Open different between Table 3 and Table 4? Are these different models?

- In Table 5, AF-AudioSet achieves comparable performance with only one-tenth of AF-Synthetic's data, suggesting dataset quality outweighs dataset scale. This observation questions the necessity of AF-Synthetic compared to AF-AudioSet.

- Comparing baseline models versus ETTA using the same training dataset (e.g., all trained on AF-Synthetic or AudioCaps) would better validate ETTA's design choices.

- The reason for not reporting some human scores in Tables 2 and 3 is unclear. Table 2 reports human scores for both ETTA and ETTA-FT-AC-100k, while Table 3 only includes ETTA.

**Methods And Evaluation Criteria:**

- The rationale behind adding each component is unclear. Additionally, comparisons seem unfair when datasets differ. For instance, the authors should provide ablations modifying each design choice while training on the same dataset as in Table 4. Currently, all models trained with the new large-scale dataset potentially mask the true effects of added modules relative to the original Stable Audio Open.

**Other Comments Or Suggestions:**

- In Section 3.1, "Table 2" should be denoted as "Figure 2."

**Other Strengths And Weaknesses:**

- Limitations have not been discussed in this paper.

**Questions For Authors:**

- In Section 3.1, when generating synthetic captions, if all generated captions have CLAP similarity below the threshold, is the audio discarded?

**Relation To Broader Scientific Literature:**

No relation to broader scientific literature is identified.

**Theoretical Claims:**

No theoretical claims are made in the main paper.

---

> ### Author Rebuttal · Authors · 2025-03-31
>
> Thank you for your review. We address your concerns as follows:
>
> **Q: Table 5 indicates that performance with AF-AudioSet is comparable or even superior… It remains challenging to conclude that AF-Synthetic is crucial for model improvement… This observation questions the necessity of AF-Synthetic compared to AF-AudioSet.**
>
> A: We showed that a scaled-up dataset with AF-Synthetic shows much better overall results for out-of-distribution data (Table 23). Note that AudioCaps and MusicCaps are considered in-distribution as both are derived from AudioSet. This means that using larger-scale training data (AF-Synthetic) provides better OOD generalizability.
>
> AF-Synthetic brings other practical benefits during training as well: we found that training ETTA using AF-AudioSet diverged after 250k training steps, whereas AF-Synthetic provides stable training up to 1M steps without instabilities. This means AF-Synthetic helps ETTA converge better from its scale.
>
> We added the training loss curves of AF-AudioSet vs. AF-Synthetic (Figure-Rebuttal-3) along with the generated samples using the OOD challenging captions to the demo webpage, to better showcase the necessity of AF-Synthetic for generalization. Link: https://anonymous.4open.science/r/etta_demo-778A/index.md
>
> **Q: The rationale behind adding each component is unclear… Additional analyses—such as human perception studies, loss curves, or convergence times—would better illustrate each component's impact.**
>
> A: To address this concern, we added loss curves to the updated demo webpage to better illustrate the design choices: https://anonymous.4open.science/r/etta_demo-778A/index.md
>
> Figure-Rebuttal-1: Stable Audio-DiT vs. ETTA-DiT, under the same v-diffusion objective and AF-Synthetic training dataset for both. We find that Stable Audio-DiT plateaus around 300K steps and starts to diverge early. ETTA-DiT continues to improve its quality with better loss for the same training steps. This shows clear benefits of the ETTA-DiT architecture.
>
> Figure-Rebuttal-2: v-diffusion vs. OT-CFM training objective, under the same ETTA-DiT architecture and AF-Synthetic dataset for both. Note that the loss scale of OT-CFM is different to v-diffusion. For prolonged training (e.g. over 500K steps), v-diffusion starts to be unstable, whereas OT-CFM provides better stability up to 1M steps. This shows practical advantages of OT-CFM over v-diffusion and is a motivation to use it for ETTA.
>
> Figure-Rebuttal-3:  AF-AudioSet vs. AF-Synthetic training dataset, under the same OT-CFM objective and ETTA-DiT architecture. AF-AudioSet quickly diverges around 250K steps and is unable to continue its training, whereas AF-Synthetic provides better convergence with continued improvements up to 1M steps.
>
> We believe these additional loss plots better illustrate the rationale behind each of the design choices of ETTA.
>
> **Q: Why are the results of Stable Audio Open different between Table 3 and Table 4? Are these different models?**
>
> A: Table 4 turned off the classifier free guidance (CFG) for all configs as an ablation study, including the original Stable Audio Open, so the difference compared to Table 3 (which uses a default CFG scale of the baselines) is expected.
>
> **Q: Table 2 reports human scores for both ETTA and ETTA-FT-AC-100k, while Table 3 only includes ETTA.**
>
> A: Thank you for mentioning this, we indeed measured ETTA-FT-AC-100k on musiccaps as well: OVL: 3.30 ± 0.10, REL: 3.44 ± 0.12. This shows that fine-tuning ETTA on AudioCaps (general sound) degrades the music generation quality in subjective evaluation, as expected. We will improve the writing to avoid confusion.
>
> **Q: In Section 3.1, when generating synthetic captions, if all generated captions have CLAP similarity below the threshold, is the audio discarded?**
>
> A: Yes, the audio is discarded in that case.

---

### Official Review · Reviewer_8RJt · 2025-03-14

**Overall Recommendation:** 4

**Summary:**

The paper provide an extensive analysis on the design choices of TTA models, achieving much superior quantitative performance to baselines across most metrics. The authors provided extensive results showing the superiority and generalization of their method as well as extensive ablations justifying their choices. Additionally, the paper present a large scale synthetic dataset which the authors has verified its effectiveness in TTA generation.

**Claims And Evidence:**

While the paper does not propose any methodological components, the authors supported all of their claims on the best practices to train TTA with convincing experiments.

**Essential References Not Discussed:**

The authors has discussed the most essential references in TTA, and dataset creation.

**Experimental Designs Or Analyses:**

The author analyses and experimental designs are sound.

**Methods And Evaluation Criteria:**

The authors evaluated their design choices using standard evaluation protocols of TTA on ambient sounds (AudioCaps) and music generation (MusicCaps), reporting metrics that shows the superiority of their methods in sound generation as well as text-audio alignment.

**Other Comments Or Suggestions:**

The conclusions of the papers could be summarized better in experiment sections, while leaving details on hyperparameters to the supplementaries. This could make the technical details of the papers easier grasp.

**Other Strengths And Weaknesses:**

In summary, the following are the main strength and weaknesses of the paper

**Strengths**
- The authors provided extensive analysis on the design choices of TTA generation
- The paper presents significantly improved performance in TTA generation compared with the open-sources models.
- The authors presented a large-scale synthetic dataset which will be useful to the community if released.
- The paper is well written.

**Weakness**
- Despite providing many analyses on design choices of TTA, the paper lacks any significant novel contribution to TTA.
- Some claim in the papers are not well addressed. The authors claim that AF-Synthetic is the first million-size synthetic caption dataset, while AutoCap [1] has open-sources a dataset of size 40+ millions.
- While the authors has presented significant quantitative improvements, their subjective evaluation is close on underperform baselines (e.g  Table 3, Table 22) .
- The authors has not promised the released of their checkpoints or code. Considering that the paper is focused around building high quality audio generation model by adapting the best practices, this could largely impact the level of the contribution of the paper.
- The paper lacks a clear discussion on the difference between the dataset collection pipeline of AF-Audioset and AF-Synthetic.

**Questions For Authors:**

- How much of the initial dataset is filtered with CLAP and others?
- AF-audioset performance is comparable with AF-synthetic in Table. 5, despite having a smaller size. While AF-synthetic achieves better performance in Tab. 23. I am wondering what would be the performance of AF-synthetic at a similar scale. Have the authors perform any experiments that shows the impact of data quantity over the generation quality?

**Relation To Broader Scientific Literature:**

While the paper does not propose any novel methodological components,  the authors has done extensive study on the design choices of TTA generation, showing superior performance. Such studies are missing from most prior work and is very useful to the community.

**Theoretical Claims:**

The paper does not present any theoretical claims.

---

> ### Author Rebuttal · Authors · 2025-03-31
>
> Thank you for your supportive review and appreciating our large-scale extensive study on the design choices of TTA to reach state-of-the-art results.
>
> **Q: AutoCap has open-sourced a dataset of size 40+ millions.**
>
> A: Thank you for mentioning the status of this concurrent work. We will discuss this concurrent work in the final version of the paper.
>
> **Q: Subjective evaluation is close or underperform baselines (e.g Table 3, Table 22).**
>
> A: For music, Stable Audio Open is trained on high quality music from Freesound and Free Music Archive (FMA) datasets, but many of these samples are not available as of today. In contrast, ETTA is not trained on any studio-quality music datasets (not even MusicCaps). As a result, ETTA has a slightly worse OVL (which only measures audio quality and does not consider text) on MusicCaps and SongDescriber.
>
> While only trained on synthetic captions, ETTA outperforms Stable Audio Open on the REL (audio-text relevance) score on AudioCaps and MusicCaps (Table 2-3), and matches it on an OOD testset SongDescriber (Table 22).
>
> **Q: The authors have not promised the release of their checkpoints or code.**
>
> A: We will release the model code and data preparation methods to reproduce the result of ETTA models.
>
> **Q: The paper lacks a clear discussion on the difference between the dataset collection pipeline of AF-Audioset and AF-Synthetic.**
>
> A: We scaled-up the data collection method proposed in AF-AudioSet [1], and added several filtering methods (Section 3.1 and Appendix C.1).
>
> [1] Kong, Zhifeng, et al. "Improving text-to-audio models with synthetic captions." arXiv preprint arXiv:2406.15487 (2024).
>
> **Q: How much of the initial dataset is filtered with CLAP and others?**
>
> A: The total filtering rate is about 73.5% (acceptance rate 26.5%) under a CLAP threshold of 0.45 and other filtering methods.
>
> **Q: AF-audioset performance is comparable with AF-synthetic in Table. 5, despite having a smaller size. While AF-synthetic achieves better performance in Tab. 23.**
>
> A: This is a great question. First of all, we find a scaled-up dataset with AF-Synthetic shows much better results for out-of-distribution data (Table 23) (Note that AudioCaps and MusicCaps are considered in-distribution, derived from AudioSet).
>
> AF-Synthetic brings other practical benefits during training as well: we found that training ETTA using AF-AudioSet diverged after 250k training steps, whereas AF-Synthetic provides stable training up to 1M steps without instabilities.
>
> We added the training loss curves of AF-AudioSet vs. AF-Synthetic along with the generated samples using the OOD challenging captions to the demo webpage, to better showcase the necessity of AF-Synthetic for generalization. Link: https://anonymous.4open.science/r/etta_demo-778A/index.md

---

### Decision · Program_Chairs · 2025-05-01

**Decision:**

Accept (poster)

**Comment:**

All reviewers recommend accepting the paper (3 accept and 1 weak accept).

The rebuttal did a good job of addressing the concerns that the reviewers had.
One reviewer (okEU) raised the score.
While all the reviewers mentioned the lack of novelty in the method, they value comprehensive large-scale experiments.
Also, the release of the checkpoints or code would be a valuable asset for the community.

The AC thus follows this unanimous recommendation by the reviewers and recommends accepting the paper.